# Treatment with galectin-1 improves myogenic potential and membrane repair in dysferlin-deficient models

**Mary L. Vallecillo-Zúniga[1], Matthew F. Rathgeber[1], P. Daniel Poulson[1], Spencer Hayes[1], Jacob S. Luddington[1], Hailie N. Gill[1], Matthew Teynor[1], Braden C. Kartchner[1], Jonard Valdoz[1], Caleb Stowell[1], Ashley R. Markham[1], Connie Arthur[2], Sean Stowell[2], Pam M. Van Ry[1]***

1 Department of Chemistry & Biochemistry, Brigham Young University, Provo, UT, United States of America,
2 Center for Apheresis, Emory Hospital, Laboratory and Blood Bank, Emory Orthopaedics and Spine Hospital, Center for Transfusion and Cellular Therapies, School of Medicine, Emory University, Atlanta, GA, United States of America

* pvanry@chem.byu.edu

**Data Availability Statement:** All relevant data are within the manuscript and its Supporting Information files except proteomic data. The

## Abstract

Limb-girdle muscular dystrophy type 2B (LGMD2B) is caused by mutations in the dysferlin gene, resulting in non-functional dysferlin, a key protein found in muscle membrane. Treatment options available for patients are chiefly palliative in nature and focus on maintaining ambulation. Our hypothesis is that galectin-1 (Gal-1), a soluble carbohydrate binding protein, increases membrane repair capacity and myogenic potential of dysferlin-deficient muscle cells and muscle fibers. To test this hypothesis, we used recombinant human galectin-1 (rHsGal-1) to treat dysferlin-deficient models. We show that rHsGal-1 treatments of 48 h-72 h promotes myogenic maturation as indicated through improvements in size, myotube alignment, myoblast migration, and membrane repair capacity in dysferlin-deficient myotubes and myofibers. Furthermore, increased membrane repair capacity of dysferlin-deficient myotubes, independent of increased myogenic maturation is apparent and co-localizes on the membrane of myotubes after a brief 10min treatment with labeled rHsGal-1. We show the carbohydrate recognition domain of Gal-1 is necessary for observed membrane repair. Improvements in membrane repair after only a 10 min rHsGal-1treatment suggest mechanical stabilization of the membrane due to interaction with glycosylated membrane bound, ECM or yet to be identified ligands through the CDR domain of Gal-1. rHsGal-1 shows calcium-independent membrane repair in dysferlin-deficient and wild-type myotubes and myofibers. Together our novel results reveal Gal-1 mediates disease pathologies through both changes in integral myogenic protein expression and mechanical membrane stabilization.

## Introduction

Limb-girdle muscular dystrophy 2B (LGMD2B) belongs to a family of muscular dystrophies called dysferlinopathies. The incidence of this disease ranges from 1:1,300 to 1:200,000, with

proteomic data that support the findings of this study are openly available in MassIVE Repository Statistic at ftp://massive.ucsd.edu/MSV000085835/

**Funding:** This study was supported by the following Brigham Young University awards: Roland K. Robbins Research Fellowship to MLVZ and Earl M. Woolley Research Innovation Award to PMVR. The authors also thank the Jain Foundation for their funding and support to PMVR.

**Competing interests:** We have read the journal's policy and authors of this manuscript have the following competing interests: The University of Nevada-Reno has been issued a patent in the U.S. (# US20130065242 A1) and Australia (# 45557BOA/VPB) for, "Methods for diagnosing, prognosing and treating muscular dystrophy". PMVR is an inventor on these patents. Strykagen currently holds the license for this technology. Brigham Young University has filed a provisional patent for "Galectin-1 immunomodulation and myogenic improvements in muscle diseases and autoimmune disorders." (#U.S. Provisional Pat. No. 62833511, Docket # 2019-015). This does not alter our adherence to PLOS ONE policies on sharing data and materials.

certain geographic locations and ethnic populations more heavily impacted than others [1–3]. Patients with this disease present muscle degeneration and weakness beginning in the second decade of life and often exhibit complete loss of ambulation by the third decade of life.

Symptoms of LGMD2B stem from mutations in the *DYSF* gene, which encodes for the dysferlin protein. Dysferlin is a 230kDa transmembrane protein heavily involved in $Ca^{2+}$ signaling in adult myocytes [4]. Mutations to the dysferlin protein lead to aberrant $Ca^{2+}$ signaling, causing poor membrane repair, myogenesis, and muscle degeneration [4–9]. Dysferlin-deficient myoblasts show decreased myogenesis, but the direct influence of dysferlin on this process is unclear [10]. Membrane repair is a complex process involving multiple pathways with the purpose of restoring compromised membrane integrity.

Current drug treatments for LGMD2B are limited and focus on mitigating the effects of chronic inflammation. Other palliative treatment options include muscle strengthening and patient education regarding preventative measures to reduce muscle injury. There is an unmet need in the field for viable long-term therapeutic options. Glucocorticoid treatments have been used to modulate impaired membrane stability and inflammatory response in many muscular dystrophies [11, 12]. However, regular glucocorticoid treatment has marginal or detrimental effects in patients with LGMD2B [11, 13]. Therefore, due to the lack of current viable treatments, a therapeutic that can increase myogenic potential and membrane repair would be most beneficial to patients and their families.

Galectin-1 (Gal-1) is a small, non-glycosylated protein encoded by the *LGALS1* gene with a carbohydrate recognition domain (CRD) which is highly conserved between all mammals with an 88% homology [14–18]. Mouse and human Gal-1 have minor structural differences, but the carbohydrate recognition residues are 100% conserved [19]. Mice lacking Gal-1 showed a reduction in myoblast fusion and muscle regeneration [20]. Recombinant human galectin-1 (rHsGal-1) has shown efficacy in reducing disease pathologies in murine models of Duchenne Muscular Dystrophy (DMD) through positive regulation of myogenesis and stabilization of the sarcolemma [15]. Since previous research using rHsGal-1 was similar to those reported using recombinant mouse Gal-1 in a DMD mouse model, we chose to use rHsGal-1 in our study. Although Duchenne Muscular Dystrophy and LGMD2B have different etiologies, they share similar pathologies such as diminished membrane repair, poor muscle regeneration, and chronic inflammation. Therefore, Gal-1 poses as an effective treatment option in increasing myogenesis and membrane repair in LGMD2B.We hypothesize that the addition of exogenous recombinant human galectin-1 (rHsGal-1) will improve myogenic regulatory factors and increase membrane repair capacity, resulting in more robust muscle formation in models of LGMD2B.

We explore the effects of rHsGal-1 treatment in A/J dysferlin-deficient ($A/J^{-/-}$) cells and *ex-vivo* muscle assessment using $Dysf^{-/-}$ (B6.129.Dysftm1Kcam/J), Bla/J (B6.A-Dysfprmd/GeneJ), and BL/6 (C57BL/6) mice. This study shows that Gal-1 treatment increases myogenic transcription factors leading to enhanced myotube formation in $A/J^{-/-}$ myotubes and increased membrane repair capacity in $A/J^{-/-}$ myotubes as well as $Dysf^{-/-}$ and WT myofibers. Additionally, this work reveals that the carbohydrate recognition domain (CRD) of Gal-1 is necessary for improved repair capacity and that the impact of Gal-1 on membrane repair is $Ca^{2+}$-independent in both diseased and non-diseased models. Together, these findings will broaden Gal-1 therapeutic applications to include LGMD2B models.

## Materials and methods

### Recombinant human Galectin-1 (rHsGal-1) production and purification

The human Galectin-1 gblock LGALS1 gene fragments were produced as doubled-stranded DNA using high fidelity polymerase (Integrated DNA Technologies, Coralville, IA). The

LGALS1 gblock was cloned into the pET29b (+) vector (kindly provided by Dr. James Moody) using NEBuilderⓇ HiFi DNA Assembly Cloning Kit (E552OS New England Biolabs (NEB), Ipswich, MA). The product was purified following the E.Z.N.A.Ⓡ Plasmid DNA Mini Kit I protocol (Omega Bio-Tek, Inc. Norcross, GA) and the DNA sequence was confirmed by Eton-Bioscience, Inc. (San Diego, CA). The cloned vector was transformed into BL21(DE3) competent E. coli cells (High Efficiency, NEB # C2527H) grown and induced with 0.1mM IPTG (Invitrogen). rHsGal-1 was purified using the Cobalt Talon Metal Affinity Resin protocol (Takara Bio USA, Inc, Mountain View, CA) in a poly-prepⓇ Chromatography column (Cat # 731–15550, Bio-Rad, Richmond, CA) and imidazole elution buffer. Purified rHsGal-1 was then filtered and dialyzed three times for a total of 24h in PBS at 4°C. Endotoxin levels were measured using LAL Chromogenic Endotoxin Quantitation Kit (Cat # 88282, Thermo Scientific Rockford, IL). All endotoxin levels of purified rHsGal-l were below the FDA limit of 0.5EU/ml at >0.1EU/ml. Purified rHsGal-1 was conjugated with Alexa Fluor 647 following the protocol provided with the protein labeling kit (Molecular Probes, Cat No. A20173, Eugene, OR) with a few alterations as described in Stowell et al. [21]. The concentration of both rHsGAL-1 and Alexa Fluor 647 labeled rHsGal-1was determined with the Pierce™ BCA Protein Assay Kit (ThermoScientific, Cat. No. 23225, Rockford, IL).

## Cell culture

Immortalized murine myoblasts H2K A/J$^{-/-}$, [A/J$^{-/-}$], (Clone #13–1,10/28/09) and H2K A/J WT, [WT], (Clone #16, 6/9.2010), kindly provided by Dr. Terence A. Partridge (Center for Genetic Medicine research, Children's National Health System, WA, DC) were cultured as described in Morgan.et al. [22] and Jat et al. [23]. Myoblasts were then plated onto glass-bottomed, collagen coated dishes sterilized with gamma-irradiation (MatTek, Part No: P35GCOL-1.0-14-C, Ashland, MA), seeded at a density of 5,555 cells/cm$^2$ and incubated at 33°C in 10% $CO_2$. Myotubes were obtained from confluent myoblasts after 2 to 4 days in differentiation media supplemented with or without rHsGal-1 (0.014μM-0.22μM). Differentiation media and treatments were changed every other day.

## Western blotting

Myotubes (at 2 to 4 days) were obtained as described above. Whole cell lysates were prepared using RIPA lysis buffer (10 mM Tris-Cl (pH 8.0), 1 mM EDTA, 1% Triton X-100, 0.1% sodium deoxycholate, 0.1% SDS, 140 mM NaCl, and 1 mM PMSF) and Halt™ Protease and Phosphatase Inhibitor Single-Use Cocktail (100X) (Cat No. 78442, ThermoScientific). Protein concentration was determined using the Pierce™ BCA Protein Assay Kit (ThermoScientific). Proteins samples were separated under standard conditions on 6%-20% SDS-PAGE gels and transferred onto Nitrocellulose Membranes 0.2 μm (Bio-Rad, Cat No.1620150, Hercules, CA) through electro blotting. After blocking with 5% w/v non-fat dry milk in 1X TBST), membranes were probed overnight for the following mouse, rabbit, or goat monoclonal and polyclonal antibodies: 6x-His Tag Monoclonal Antibody (HIS.H8), (Cat. No. 14-6657-80 Invitrogen, 1:1000), Galectin-1 Monoclonal Antibody (6C8.4–1) (Cat. No. 43–7400, Invitrogen 1:1000), Myogenin (FD5, DSHB, 0.2 μg/mL, Pax7 (DSHB, 0.2 μg/mL), Myf5(Cat. No. PA5-47565, Invitrogen, 1.5 μg/mL), MyoD (5.8A, ThermoScientific, 2.5 μg/mL), MHC (MF20, myosin, sarcomere, Cat. No. AB_2147781, DSHB, 0.2 μg/mL), Annexin A6 (Cat No. 720161, ThermoFisher Scientific, 2 μl), Annexin A1 (Cat. No. 713400, Invitrogen, 1:1000), β-Tubulin Loading Control, BT7R, (Cat. No. MA5-16308, ThermoScientific, 1: 2,000), GAPDH (Cat. No. MA5-15738, Invitrogen, Rockford, IL, 1:1000), and Anti-β-actin (Cat. No. A5441, Sigma-Aldrich, St. Louis, MO. 1: 5,000). After washing primary antibodies, blots were probed using

the following secondary antibodies IRDye® 800CW Donkey Anti-Rabbit IgG (H + L) (Cat No. 926–3221, Licor, Lincoln, NE, 1: 15,000), Goat anti-Mouse IgG (H+L) Highly Cross-Adsorbed Secondary Antibody, Alexa Fluor Plus 800 (Cat No. A-32730, Invitrogen, 1: 40,000), Goat anti-Mouse IgG (H+L) Highly Cross-Adsorbed Secondary Antibody, Alexa Fluor 680 (Cat. No. A-21058, Invitrogen, 1: 5,000), and IRDye® 680RD Donkey Anti-Goat IgG (Cat. No. 926–68074, Licor, 1:5000). The blots were developed using the Odyssey CLx (Model No. 9140, Li-Cor, Lincoln, NE). Quantifications were done by using ImageJ as described in Schindelin et al. [24].

## Immunofluorescence

A/J$^{-/-}$ and A/J WT myotubes cultured onto 35 mm Glass Bottom Microwell Dishes (Cat. No. P35GCol-1.0-14-C, MatTek, Ashland, MA) were fixed in 4% paraformaldehyde, permeabilized in 0.1% triton X-100 (in PBS), and blocked using MOM IgG blocking solution for 1 h at room temperature. The myotubes were then incubated overnight at 4˚C with the appropriate primary antibody: Alexa Fluor 647/Phalloidin (Cat No. A2287, Invitrogen, 1:50), Myf5(Cat No. PA5-47565, Invitrogen, 5μg/ml), MHC (MF20, myosin, sarcomere, Cat. No. AB_2147781, DSHB 2μg/ml, DSHB), CellBrite™ Cytoplasmic Membrane Dyes (Cat No. 30021, Biotium, Fremont, CA, 5 μ/ml). Nuclei were counterstained with Hoeschst 33342 (Cat No. 62249, Thermo-Scientific, 1 μg/ml) and 4',6-diamindino-2-phenylindole (DAPI) (Cat No. 62248, Thermo Scientific, 1:500). Blots were probed using the following secondary antibodies: Fluorescein (FITC) AffiniPure Rabbit Anti-Goat IgG, Fc fragment specific (Cat. No. 305-095-046, Jackson Immune Research laboratory, West Grove, PA, 1:50), Goat anti-Mouse IgG (H+L) Highly Cross-Adsorbed Secondary Antibody, Alexa Fluor Plus 488 (Cat. No. A32723, ThermoFisher, 10 μg/ml), Goat anti-Mouse IgG (H+L) Highly Cross-Adsorbed Secondary Antibody, Alexa Fluor 680 (Cat. No. A21058, ThermoFisher, 10 μg/ml). Myotubes were mounted on coverslips using ProLong™ Diamond Antifade Mountant (Cat No. P36965, Invitrogen) and dried overnight. Images were taken on the A TCS SP2 two-photon confocal scanning microscope with LASX imaging software (Leica Microsystems Inc., Buffalo Grove, IL). 647rHsGal-1 inside-outside fluorescent values were obtained as described in Fitzpatrick et al. [25]. Inside-outside ratio was calculated by averaging three ROI from inside a cell and three ROI between cells per image.

## Fusion index scoring

A/J$^{-/-}$ and A/J WT myoblasts were plated onto in 35 mm Glass Bottom Microwell Dishes (Cat. No. P35GCol-1.0-14-C, MatTek, Ashland, MA). At 80%–90% confluence, myoblasts were differentiated as described above and were given treatment (0.11 μM rHsGal-1) or not. After three days in differentiation media and treatment, myotubes were fixed, permeabilized, stained and imaged as described above. Fusion index was calculated as the number of nuclei contained within myotubes (cells were considered to be myotubes if they contained three or more nuclei) divided by total number of nuclei. Minimum Feret's Diameter (MFD) was calculated by using ImageJ. Myotubes were outlined using the polygon tool, after which the MFD was calculated with the Feret's Diameter plugin. Alignment was calculated as described [26].

## Migration assay

12-well plates (Cat. No. 83.3921.300, SARSTEDT, Newton, NC) were prepared by placing a silicone insert (Cat. No. 80209, ibidi culture insert 2 well, Martinsried, Germany) in the center of each well. A suspension of 145,000 cells/ml (either WT and A/J$^{-/-}$ myoblasts) was prepared in growth media as described above and 70 μl of the suspension was placed into each side of the

insert. After 2 days, cells were placed in normal differentiation media or differentiation media supplemented with 0.11 μM rHsGal-1 and incubated for 2 days. To form the wound, the silicone insert was removed 1 h prior to first image after washing with PBS; Rate of migration was calculated over a 48 h period. Differentiation media or differentiation media supplemented with 0.11 μM rHsGal-1 was then replaced as described above and directly placed into the Incucyte®. Magnification was set to 10x and images were taken every 3 h for 48 h. Images were analyzed with ImageJ [24].

## Laser injury assay

A/J WT and A/J$^{-/-}$ 0.11 μM rHsGal-1 treated or NT myotubes were prepared for laser injury as described above in 35 mm Glass Bottom Microwell Dishes (Cat. No. P35GCol-1.0-14-C, Mat-Tek, Ashland, MA). After washing with PBS, the myotubes were incubated for 10 min in PBS enriched with or without: 1mM Ca$^{2+}$ (as CaCl$_2$), 1 μM intracellular (1,2-Bis(2-aminophenoxy) ethane-N, N, N′, N′-tetraacetic acid tetrakis (acetoxymethyl ester); (BAPTA-AM) (Cat. No. 196419-25MG, EMD Millipore), DMSO as a vehicle (Cat. No. 67-68-5, EMD Millipore), 1 μM (ethylene glycol-bis(β-aminoethyl ether)-N,N,N′,N′-tetraacetic acid; (EGTA) (Cat. No. 409910250, Acros Organics), 20 mM lactose (Cat. No. A11074, Alfa Aesar) or 20 mM sucrose (Cat. No. 57-50-1, Carolina Biological), and 2.5 μM FM™ 1–43 dye (N-(3-Triethylammonium-propyl)-4-(4-(Dibutylamino) Styryl) Pyridinium Dibromide)3,5 (ThermoScientific, Cat. No. T35356, Waltham, MA) for 5 min before injury. A TCS SP2 two-photon confocal scanning microscope (Leica) was used to injure the membrane of a myotube or myofiber and images were taken before and after the injury event. Pre-injury images depict uninjured myofibers. Myoblasts were not used in injury protocols, only cells with greater than 3 nuclei were counted as myotubes. The myotube was injured with a 405 nm UV laser at 100% power on a HCX PL APO CS 63.0 x 1.40 oil-objective lens. Post-injury images were taken every 5 sec for a total of 150 sec. Specific settings used as described in Carmeille et al. [27]. At least three different myotubes were selected to be injured in each dish. The total change in fluorescence intensity of FM™ 1–43 dye at the site of the wound for each time point relative to the pre-injury fluorescent intensity was measured using ImageJ [24].

## Muscle fiber isolation

A 12-well plate (Cat. No. 665 180, Grenier Bio-One) was prepared as described in Demonreun et al. [28]. After preparation of digestion plate, C57B6 and Dysf$^{-/-}$ (B6.129-Dysf$^{tm1Kcam}$/J) mice were euthanized in accordance with Brigham Young University-approved protocol. When the mice were sacrificed, hind limbs were removed and the tibialis anterior, flexor digitorum brevis, and/or gastrocnemius were excised. Next, by using a small-bore pipette, the fibers were transferred to in 35 mm Glass Bottom Microwell Dishes and allowed to attach for at least 15 min. Fibers were then treated or not with 0.11 μM rHsGal-1 and kept at 37˚C until injury. At least three different myofibers in each dish were selected to be injured. The total change in fluorescence intensity of FM™ 1–43 dye at the site of the wound for each time point relative to the pre-injury fluorescent intensity was measured using ImageJ [24].

## Quantitative RT-PCR

Total RNA was isolated from 3 days differentiated A/J WT, A/J$^{-/-}$, and A/J$^{-/-}$ treated with 0.11 μM rHsGal-1 myotubes (n = 6 independent clonal lines for each treatment group) using Quick-RNA™ Miniprep kit (ZYMO Research, Irvine, CA). Isolated RNA was reverse transcribed using SuperScript™ IV VILO™ (ThermoFisher) following the manufacturer's instructions. Real-time analysis was performed on an Applied Biosystems® QuantStudio® 5 Real-

Time PCR System using TaqMan® Fast Advanced Master Mix and TaqMan® Assays. Relative gene expression levels and statistical significance were calculated by normalizing raw Ct values to 18S, and then by using the ΔΔCt method with Applied Biosystems™ Relative Quantitation Analysis Module software [29].

## Shotgun proteomics

AJ WT, AJ$^{-/-}$, and AJ$^{-/-}$ myotubes treated or not with 0.11 μM rHsGal-1 were prepared as described in Wiśniewski et al. [30] with some alterations: Tris/HCl pH 8.5. Protein concentration were analyzed by Peirce™ BCA Protein assay kit (Thermo Scientific™) following manufacturer's instructions. Protein digestion was initiated by adding Peirce™ MS grade trypsin protease (Thermo Sceintific™ cat. # 90058) to a 1:50 μg ratio and incubated by shaking at 37˚C overnight. Digestion was quenched by 100 mM phenylmethanesulfylfluoride (PMSF, CAS # 329-98-6) in ethanol (final concentration 1 mM) and centrifuged at 14,000g for 30min. The filtrate was transferred to Thermo Fisher™ 11 mm mass spectrometry vial. Samples went through vacuum centrifugation and resuspended in orbitrap run buffer. Sample analyzation was conducted on Thermo Fisher™ Q-Exactive Obitrap. Gene Ontology analysis was processed through Princeton University Lewis-Sanger Institute for Integrative Genomics Term Finder.[31] Raw proteomic data can be found at doi: 10.25345/C5816M.

## Kinetic proteomics

A/J-/- and A/J cells were cultured at 33˚C and 10% CO2 in a growth media consisting of 20% FBS, DMEM Corning Ref #10-013CVR, 1%P/S, 2% Chicken Embryo Extract, 2 uL/mL of γ-IFN. A/J-/- and A/J cells were cultured at 37˚C at 5% CO2 in a differentiation media consisting of 5% Horse Serum, 1% P/S DMEM Corning Ref #10-013CVR. Cells were plated at a density of 100,000 cells/mL in growth media until 80–90% confluency (1–2 days). Plates were washed with PBS and changed into differentiation media, along with D2O and rHsGal-1 treatment if needed, as myotubes formed (3–4 days) with media and treatment being changed every other day. Myotubes were scraped at 3, 5, 7, 14, and 28 days. Myotubes were added into 0.5 mL of media and pelleted at 1200 rpm for 5 min, flash frozen and then kept at –20˚C for one day and then moved to –80˚C storage. Myotubes and the isolated proteins were prepared as described above for proteomic analysis. Turnover rates were calculated as previously described in Price et al. [56]. Accession numbers were identified for each protein of interest, and their relative abundance and turnover rates were aggregated. Turnover graphs were made under a single-pooled rise to plateau model.

## Statistical analysis

Data analysis were completed by using Tukey's multiple comparison test 1-way and 2-way ANOVA, the Student's t test, Welch's, and Bartlett's test through the GraphPad Prism Software version 8.0. For membrane repair analysis, the data are conferred the averaged values for all the myotubes used in the analysis, and the treatment at individual time points. P values are indicated in the figure when statistical significance is determined for all groups as described in the figure legends.

# Results

## Production and purification of rHsGal-1

Gal-1 induces skeletal muscle differentiation and decreases disease manifestation in DMD [15, 32]. Exogenous Gal-1 may positively modulate different pathologies in LGMD2B. To explore

the effects of Gal-1 treatment in *Dysf*-deficient models, endotoxin-free rHsGal-1 was produced using the pET29b(+) vector with a C-terminal 6X Histidine tag for easy detection during purification and expression steps. Purification and detection analyses were made by total protein stains and western blot (S1A–S1E Fig).

## rHsGal-1 increases myogenic potential in A/J$^{-/-}$ myotubes

The formation of myotubes is a multi-step process incorporating migration, adhesion, and alignment, followed by formation of extracellular proteins that coordinate cellular stability [33]. Gal-1 expression levels during myoblast growth, differentiation and repair play a key role in forming healthy skeletal muscle. The lack of Gal-1 leads to poor myotube formation and delays in myoblast fusion [20, 34].

Myogenin is a muscle-specific transcription factor expressed by terminally differentiated myotubes and is known to be decreased in immortalized A/J$^{-/-}$ myotubes [35]. However, after a 72 h treatment with rHsGal-1, myogenin expression increased in A/J$^{-/-}$ myotubes (Fig 1A

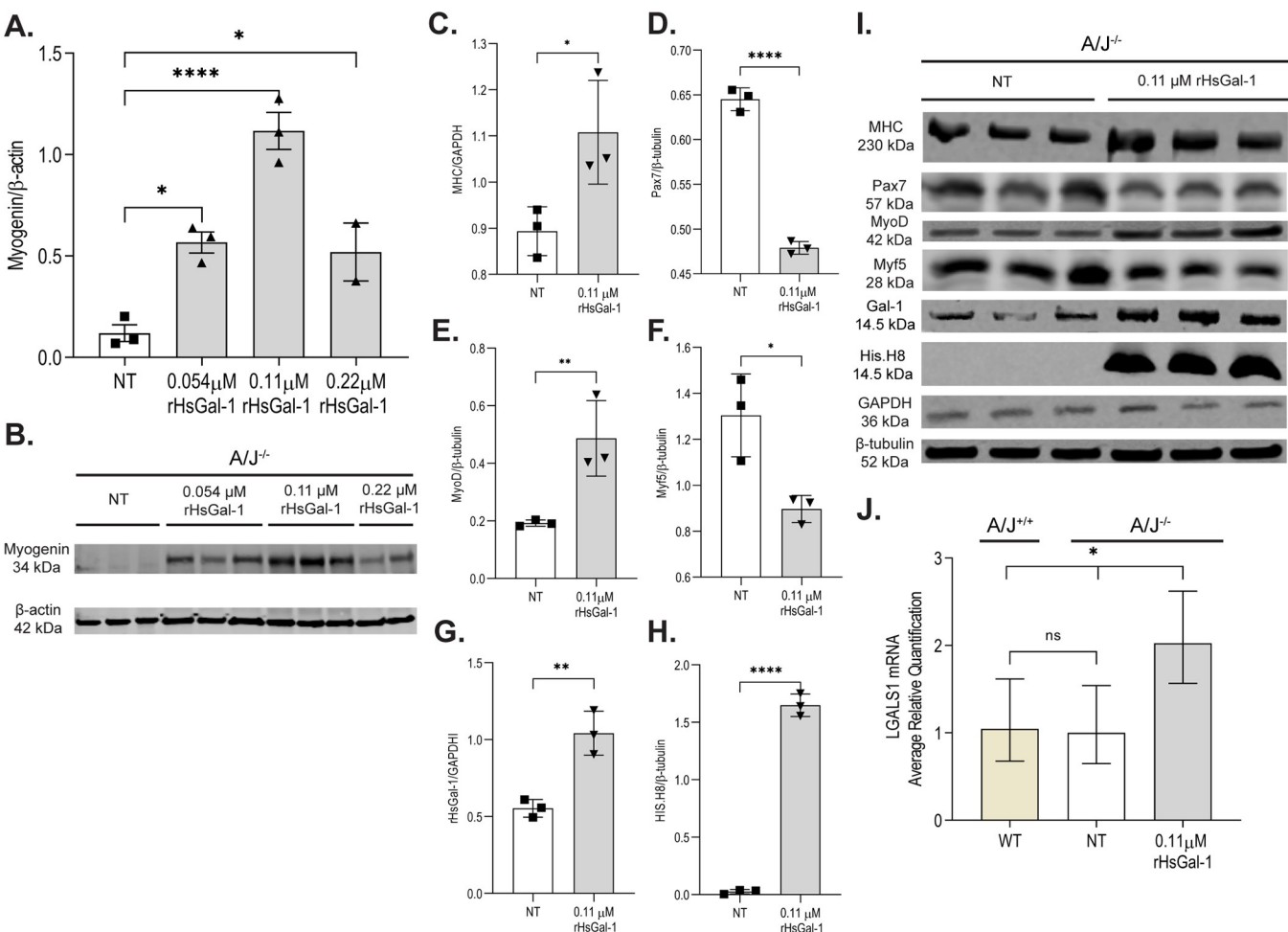

**Fig 1. rHsGal-1 increases myogenic regulatory factors in A/J$^{-/-}$ myotubes.** A. Quantification of myogenin after 72h treatment with varying concentrations of rHsGal-1. B. Western blot images of myogenin at different rHsGal-1 treatments. C-F. Quantification of myogenic markers MHC(C), Pax7(D), MyoD(E), and Myf5(F) in A/J$^{-/-}$ myotubes after 72h treatment with 0.11μM rHsGal-1. G-H. Quantification of Gal-1(G) and His.H8(H)in A/J$^{-/-}$ myotubes after 72h treatment with 0.11μM rHsGal-1. I. Western blot images of myogenic markers (Pax7, Myf5, MyoD, and MHC) and of mouse Gal-1 and His Tagged rHsGal-1. J. RT-qPCR quantification of LGALS1 transcript between A/J WT, A/J $^{-/-}$ NT, and A/J $^{-/-}$ 0.11μM rHsGal-1 treated myotubes. p values are measured by Tukey's multiple comparison test and indicated by *p< 0.05, **p< 0.01, ****p< 0.0001 (n = 3 for each group). Error bars represent SEM.

and 1B). To determine the most efficacious dose of rHsGal-1 required to increase myogenesis, A/J$^{-/-}$ myoblasts either received no treatment (NT) or were treated with three concentrations of rHsGal-1 for 72 h post differentiation. When compared to NT myoblasts our results show a 4.74-, 9.35- and 4.35-fold increase in myogenin with 0.054 μM, 0.11 μM, and 0.22 μM rHsGal-1 treatment, respectively (Fig 1A and 1B). To further investigate changes in myogenic potential, we examined whole cell lysates and measured levels of early, mid, and late myogenic markers: paired box protein 7 (Pax7), myogenic factor 5 (Myf5), myoblast determination protein (MyoD) and myosin heavy chain (MHC) respectively, after treatment with a dose of 0.11 μM rHsGal-1 (Fig 1C–1F). Levels of early stage markers decreased, while levels of late stage markers increased by 2.5-fold (MyoD) and 1.46-fold (MHC) when treated with 0.11 μM rHsGal-1 (Fig 1C–1F and 1I). The removal of 0.11 μM rHsGal-1 after a 10 min treatment in A/J$^{-/-}$ myotubes followed by 72 hours in differentiation media show no significant difference in Myf5 or MHC expression when compared to NT (S2A–S2C Fig). The changes in myogenic transcription factors were validated using immunofluorescent imaging. After 48 h of differentiation, early myotube populations with or without treatment were stained with a nuclear counterstain 4′,6-Diamidino-2-Phenylindole, dihydrochloride (DAPI), an actin filament stain (Phalloidin) and anti-Myf5. Images reveal that there was no Myf5 visible in WT or rHsGal-1 A/J$^{-/-}$ treated myotubes, while Myf5 positive myoblasts are observed in A/J$^{-/-}$ NT (Fig 2C).

In order to show that rHsGal-1 treatment was the cause, we investigated transcript and protein levels of Gal-1. RT-qPCR analysis revealed *LGALS1* mRNA transcript levels were doubled after a 72h 0.11μM rHsGal-1 treatment post differentiation (Fig 1J). Increases in rHsGal-1, 6XHis-tag protein, and *LGALS1* mRNA transcripts levels correlate with rHsGal-1 treatment and suggest a positive feedback loop that ultimately upregulates myogenic transcription factors in diseased cells with a 72h treatment (Fig 1G–1J). The levels of 6XHis-tag after 72 hours also indicate that the exogenous Gal-1 is internalized and stable within cell culture.

Gal-1 knockout mice are reported to have decreased myofiber formation [20]. We explored the ability of rHsGal-1 to increase fusion capacity of A/J$^{-/-}$ myotubes by measuring fusion index, alignment and size. Dysferlin-deficient myotubes were stained with Phalloidin or MHC and DAPI in order to determine fusion index (Fig 2A and 2B). Treatment with 0.11μM rHsGal-1 showed a dramatic increase in number of nuclei per myotube (WT = 11.4±0.59, NT = 6.43 ±0.37, and rHsGal-1 = 14.5±0.65) (Fig 2D) and average fusion index (WT = 0.90±0.003, NT = 0.85±0.007, rHsGal-1 = 0.96± 0.004) (Fig 2E). Myotube and myofiber alignment have been shown to lead to improved muscle development and strength [36]. rHsGal-1 treatment led to significantly improved myotube self-alignment compared to WT and NT myotubes (Fig 2F). Additionally, 0.11μM rHsGal-1 treatment myotubes were 25% larger than WT and 35% larger than NT. These data provide further evidence that rHsGal-1 treatment increases fusion capacity of A/J$^{-/-}$ myoblasts towards formation of myotubes (Fig 2G). An *in vitro* migration assay showed increased myoblast migration and *de novo* myotube formation within the injured area. Our treatment groups had the following average wound closure rates: NT = 1.34 ± 0.12% area/hour, WT = 1.69 ± 0.058% area/hour and 0.11μM rHsGal-1 = 1.77 ± 0.063% area/hour (Fig 2H). This set of experiments suggest that low doses 48h treatment of rHsGal-1 in an *in vitro* dysferlin-deficient model may increase myogenic potential in myoblasts.

## Increased rHsGal-1-mediated repair is dependent on the CRD of rHsGal-1 and independent of Ca$^{2+}$ in both dysferlin-deficient and non-diseased models

The major pathological feature in LGMD2B is compromised membrane repair. To explore the effectiveness of rHsGal-1 treatment on the membrane repair process, we employed a

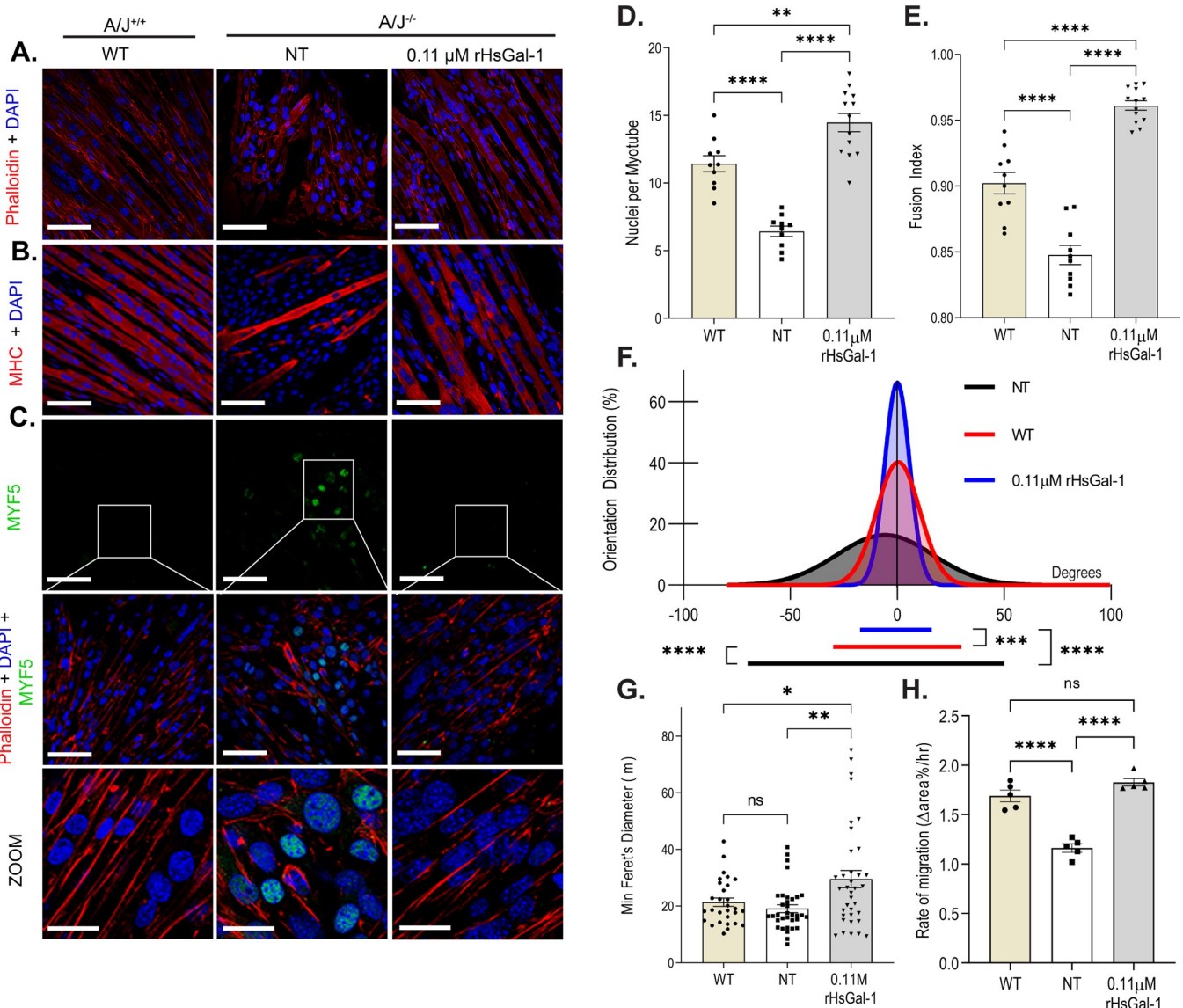

**Fig 2. rHsGal-1 treatment increases levels of fusion index and myotube maturity.** A. Representative images of A/J cells cultured and immunostained with Phalloidin (red) and DAPI (blue). B. Representative images of A/J cells cultured and immunostained with MHC (red) and DAPI. C. Representative images of A/J cells cultured and immunostained with Myf5 (green), Phalloidin, and DAPI. D. Average number of nuclei per myotube compared between WT (n = 1608 nuclei, 187 myotubes, 10 fields), NT (n = 1587 nuclei, 215 myotubes, 9 fields), and 0.11 μM rHsGal-1 treated (n = 2476 nuclei, 166 myotubes, 13 fields) groups E. Fusion index between WT, NT, and 0.11μM rHsGal-1 treated myotube groups. F. Myotube alignment along the major axis compared between WT (n = 50 myotubes, 10 fields), NT (n = 49 myotubes, 9 fields) and 0.11 μM rHsGal-1 treated (n = 75 myotubes, 13 fields) myotubes. G. Minimum Feret's diameter measurements between WT (n = 30 myotubes, 10 fields), NT (n = 34 myotubes, 9 fields), and 0.11μM rHsGal-1 treated (n = 36 myotubes, 13 fields) myotubes. H. Rate of migration between WT, NT, and 0.11μM rHsGal-1 treated myoblast groups. p values are measured by Tukey's multiple comparison test and indicated by $^{**}$p<0.01 and $^{****}$p< 0.0001. Error bars represent SEM. Scale bar = 100 μm.

membrane laser injury assay on dysferlin-deficient myotubes (myotubes defined as having >3 nuclei) in the presence of FM1-43, a lipophilic dye that fluoresces when bound to lipids. We quantified the change in fluorescence after injury; cells with less dye entry indicate better membrane repair (Fig 3A, S1 Video). These injuries were performed after 10min and 48h treatments to evaluate any time-dependencies. After laser injury, we quantified changes in fluorescent intensity to measure effectiveness and kinetics of membrane repair. At 150s post-

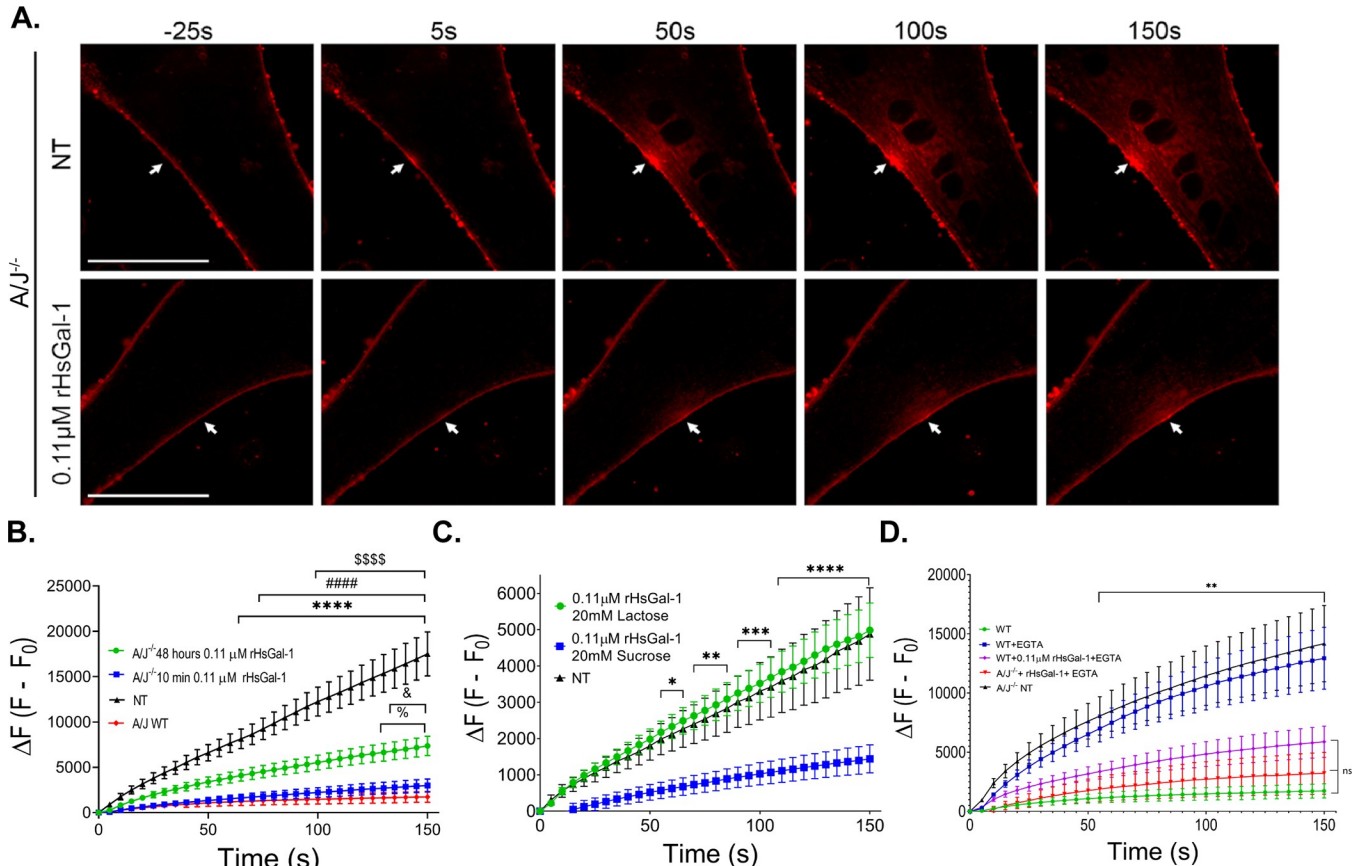

**Fig 3. rHsGal-1 treatment increases membrane repair capacity in A/J$^{-/-}$ and A/J$^{+/+}$ myotubes dependent on CRD activity.** A. Representative images of FM 1–43 dye accumulation in NT and 48h 0.11 μM rHsGal-1 treated A/J$^{-/-}$ myotubes after injury with UV laser. White arrows indicate site of injury. B. Quantification of the change in fluorescent intensity inside A/J$^{-/-}$ myotubes following laser injury when treated with 0.11μM rHsGal-1 for 10min and 48h compared to WT A/J$^{+/+}$ and NT A/J$^{-/-}$ myotubes. C. Change in the fluorescent intensity in 0.11μM rHsGal-1 treated A/J$^{-/-}$ myotubes supplemented with lactose and sucrose compared to NT A/J$^{-/-}$ myotubes. D. Change in the fluoresce intensity in 0.11μM rHsGal-1 treated A/J$^{+/+}$ myotubes supplemented with or without EGTA and rHsGal-1 compared to WT A/J$^{+/+}$ and NT A/J$^{-/-}$ myotubes. Values were measured by Tukey's multiple comparison test and indicated by in B. ****p< 0.0001 A/J$^{-/-}$ NT vs A/J WT, ####p <0.0001 between A/J$^{-/-}$ NT vs A/J$^{-/-}$ 10min rHsGal-1, \$ \$ \$ \$p<0.0001 between A/J$^{-/-}$ NT vs A/J$^{-/-}$ 48hr rHsGal-1, % p<0.05 between A/J$^{-/-}$ 48hr rHsGal-1 and A/J WT, and & p<0.05 between 10min and 48h. C. *p<0.05, **p<0.01, ***p<0.001. and ****p< 0.0001 between A/J$^{-/-}$ 0.11μM rHsGal-1 20mM sucrose vs. A/J$^{-/-}$ 0.11μM rHsGal-1 20mM lactose and A/J$^{-/-}$ NT. D. at least significance of **p<0.01 between A/J WT, AJ WT + EGTA + 0.11μM rHsGal-1, and A/J$^{-/-}$ 0.11μM rHsGal-1 + EGTA vs. A/J$^{-/-}$ + EGTA and A/J WT + EGTA Scale bars = 50 μm. Error bars represent SEM. n ≥ 11 from 2 independent experiments for each group.

injury, A/J$^{-/-}$ myotubes treated with 0.11μM rHsGal-1 for 48h had 58% less dye entry than NT, while a 10min treatment decreased final fluorescent intensity by 83% from NT. WT had 90.1% less dye entry than NT. In comparing dysferlin-deficient treatments to WT cells, after a 10min and 48h treatment A/J$^{-/-}$ cell only allowed 7% and 32% more dye than non-diseased cells (Fig 3B).

To determine the involvement of the Gal-1 carbohydrate recognition in repair capacity, we performed a laser ablation assay in the presence of lactose or sucrose. The CRD of Gal-1 is known to have a binding affinity for lactose whereas sucrose does not interact with the CRD [21, 37]. When A/J$^{-/-}$ myotubes were incubated with 20mM sucrose and 0.11μM rHsGal-1 10min prior to treatment, we observed an increase in membrane repair capacity consistent with previous results. However, when rHsGal-1 CRD interactions were inhibited with lactose, we saw no increase in membrane repair (Fig 3C). We conclude that the CRD plays a crucial role in the membrane repair mechanism of Gal-1.

Non-diseased models show that dysferlin-mediated repair is dependent on intrinsic $Ca^{2+}$ signaling properties of dysferlin [38–40]. Therefore, dysferlin-deficient muscle fibers are defective in many $Ca^{2+}$ sensitive processes, including membrane repair. We conducted a group of laser injury assays to determine the role of $Ca^2$ in rHsGal-1 mediated. Dysferlin-deficient myotubes treated with 0.11μM rHsGal-1 for 48h had a final change in fluorescent intensity 57% lower than NT A/J$^{-/-}$ myotubes 150s post injury, independent of the presence of $Ca^{2+}$ in their cell media (S3A Fig). To better understand the $Ca^{2+}$ independent therapeutic benefit of Gal-1 in A/J$^{-/-}$ myotubes, we quantified final fluorescent intensity in the presence and absence of extracellular (EGTA) and intracellular (BAPTA-AM) calcium chelators. We saw that rHsGal-1 treatment increases membrane repair and mitigates effects of dysferlin-deficiency in the presence of both intracellular and extracellular calcium chelators (S3B and S3C Fig). Calcium imaging using Fluo-4AM also revealed no increase in $Ca^{2+}$ accumulation at site of injury in A/J$^{-/-}$ 0.11 μM rHsGal-1 treated and NT myotubes, but did find an increase in $Ca^{2+}$ accumulation at the site of injury in A/J WT myotubes (S3D–S3G Fig). Next, we wanted to determine the positive impact of rHsGal-1 on membrane repair in the presence of dysferlin through A/J$^{+/+}$ WT myotubes. We used EGTA to inhibit the normal, calcium-dependent function of dysferlin in WT A/J$^{+/+}$ myotubes. Our results showed no significant differences in membrane repair between non-treated A/J$^{-/-}$ and WT myotubes treated with EGTA. Although WT myotubes treated with EGTA showed reduced repair due to lack of extracellular $Ca^{2+}$, WT myotubes treated with 0.11 μM rHsGal-1 plus EGTA showed a significant improvement in membrane repair similar to A/J$^{-/-}$ myotubes treated with 0.11 μM rHsGal-1 (Fig 3D). Even when deprived of $Ca^{2+}$, WT cells treated with Gal-1 are able to alleviate repair defects due to lack of $Ca^{2+}$.

## *Ex-vivo* rHsGal-1 treatment increases membrane repair capacity in Dysf$^{-/-}$ myofibers

To verify *in vitro* myotube injury results, myofibers taken from *Dysf*$^{-/-}$ and Bla/J mice were treated with 0.11 μM rHsGal-1 for 2 h prior to injury. Our results showed a 70% decrease in final fluorescent intensity from NT in the *Dysf*$^{-/-}$ myofiber and a 57% decrease compared to NT in the myofiber from the Bla/J mice (Fig 4A–4C). Injured mice fibers taken from C57BL/6 (WT) mice treated with or without rHsGal-1 showed no significant differences in membrane repair (Fig 4D). Additionally, we used EGTA to inhibit the normal, calcium-dependent function of dysferlin in WT myofibers. When treated with EGTA, WT myofibers showed an increased dye entry of 50% compared to WT without EGTA. However, WT myofibers treated with EGTA plus 0.11 μM rHsGal-1 were not significantly different from WT myofibers untreated with EGTA or rHsGal-1 (Fig 4E). These *ex vivo* results give further weight to *in vitro* myotube data.

## rHsGal-1 localizes at the site of injury and sites of cellular fusion in dysferlin-deficient myotubes

We next examined temporal-spatial localization of rHsGal-1 during laser injury and during myotube formation using AlexaFluor-647 conjugated rHsGal-1 (647rHsGal-1). 647rHsGal-1 localized on the membrane of myotubes after 10min incubation (Fig 5A). However, after a 48h treatment there was minimal rHsGal-1 localized on the myotube membrane and instead formed puncta within the cytosol (Fig 5B and 5H), further indicating the stability of the exogenous Gal-1 within these cells. After laser injury in the 48hr 647rHsGal-1 treated myotubes, we observed 647rHsGal-1 concentrate at the site of injury (Fig 5B). Confluent A/J$^{-/-}$ myoblasts treated with 647rHsGal-1 in differentiation media for 10min, 4h, 8h, 24h, and 48h were imaged to resolve differences in membrane versus nuclear localization. 647rHsGal-1 in confluent myoblasts treated for 10min accumulated on the membrane and intramembrane space

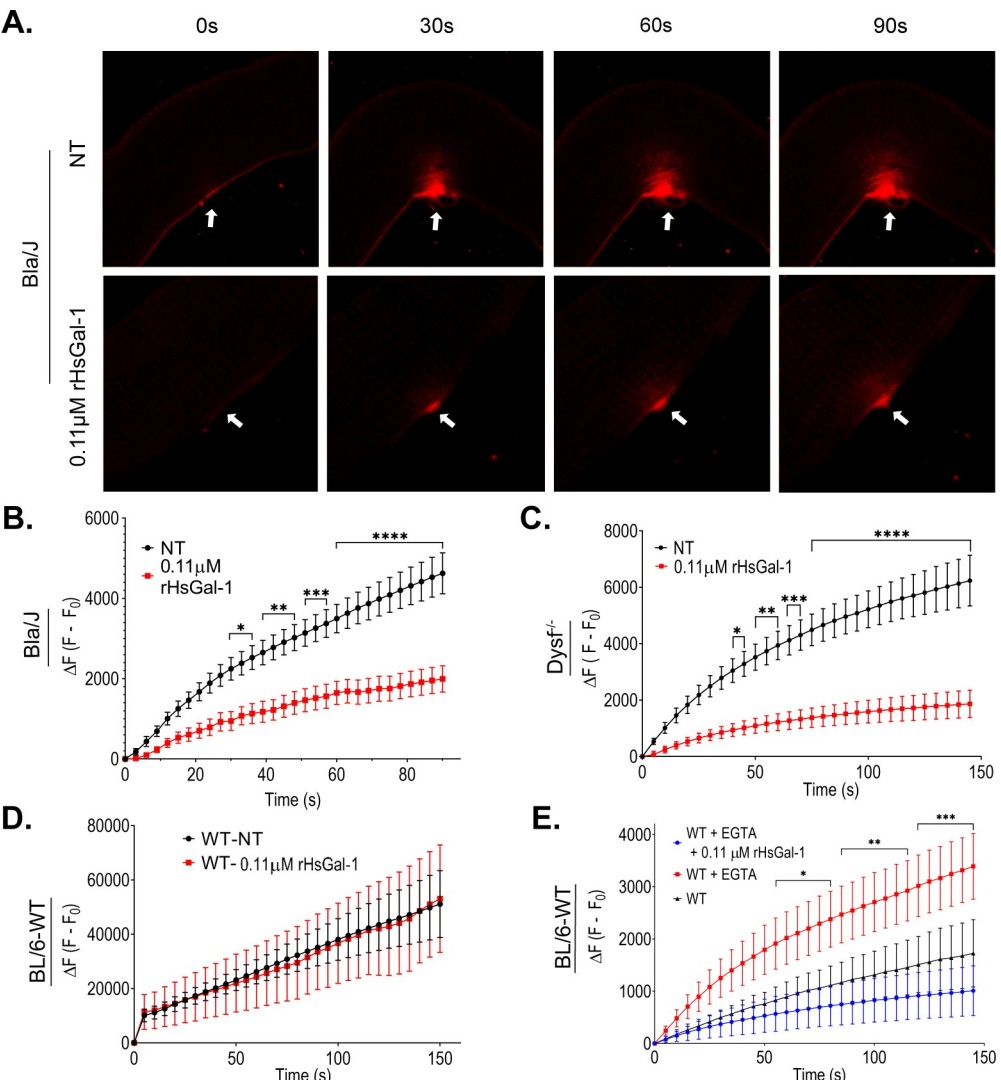

**Fig 4. rHsGal-1 treatment positively effects sarcolemma and membrane repair in a Ca²⁺ depleted environment.** A. Representative images at time points 0s, 30s, 60s, 90s of FM1-43 dye accumulation in Bla/J mouse fibers upon laser injury with a 405nm laser. White arrows indicate site of injury. B. Quantification of the total change of fluorescence in Bla/J myofibers. C. Quantification of the total change of fluorescence in *Dysf⁻/⁻* myofibers post injury. D. Quantification of the change in fluorescence in C57BL/6-WT myofibers after treatment with or without rHsGal-1. E. Quantification of the change in fluorescence in C57BL/6-WT myofibers after treatment with or without EGTA and rHsGal-1. p values were measured by Tukey's multiple comparison test and indicated by *p< 0.05, **p< 0.01, and ***p< 0.001. Scale bars = 50 μm. Error bars represent SEM. n≥ 15 myofibers per condition.

(Fig 5D). By 4 h of treatment, 647rHsGal-1 dispersed throughout the intracellular and extra-cellular space (Fig 5E). Beginning at 4 h and 8 h, 647rHsGal-1 appears to coalesce in the shape of an extracellular lattice (Fig 5E and 5F, S2 Video) which expands in both 24h and 48 h images (Fig 5G and 5H). 48h post-treatment, we saw mature myotubes with intracellular rHsGal-1 and extracellular lattice structures of rHsGal-1 at sites of cellular fusion (Fig 5G). Quantifica-tion of our results show after 4 h treatment 647rHsGal-1 is predominately located inside myo-blasts but by 8 h and beyond most of the rHsGal-1 is found outside the cells (Fig 5C). Additionally, we saw 647rHsGal-1 encapsulated in lipid layers, suggesting the formation of vesicles (Fig 5I, S3 Video).

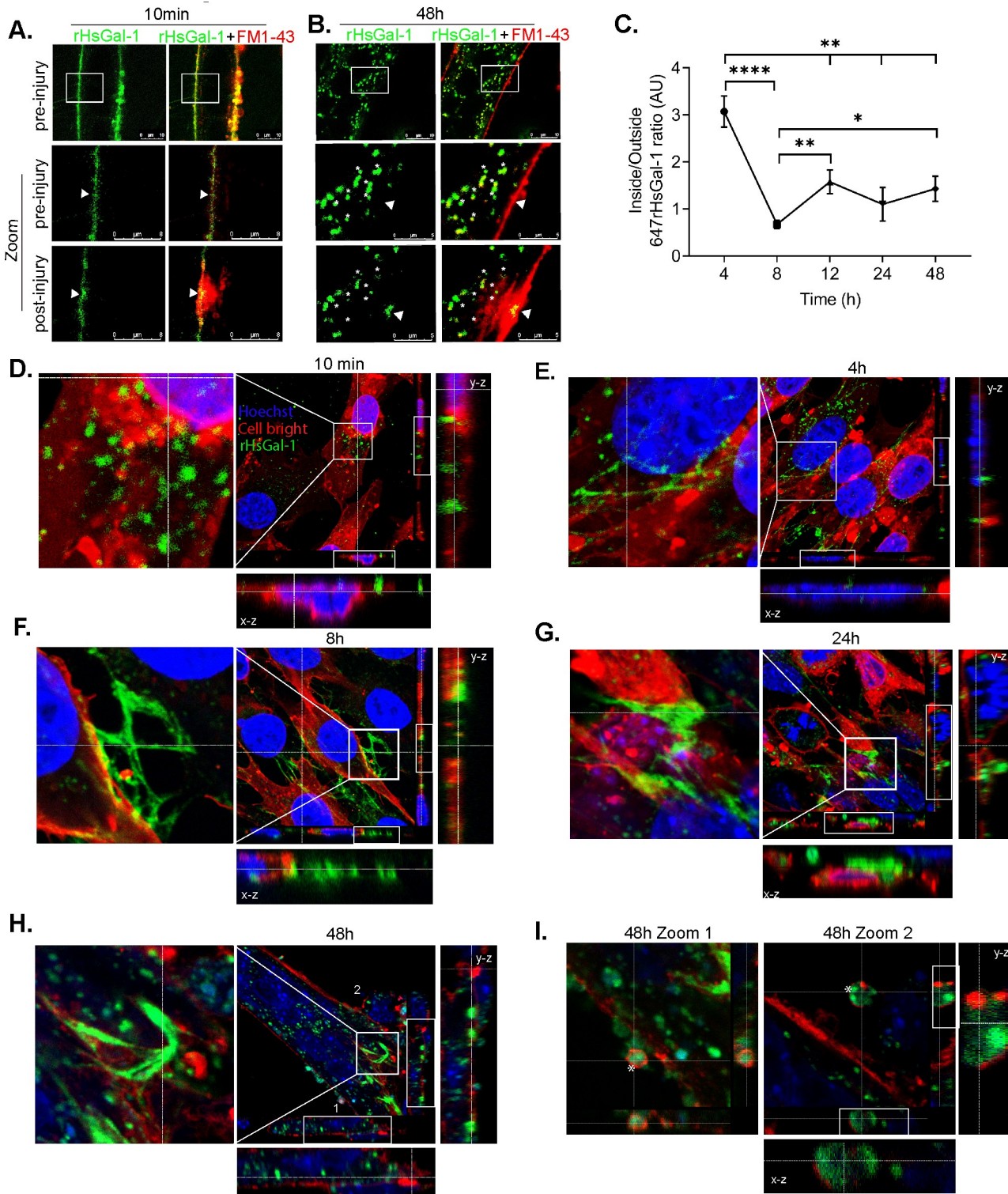

**Fig 5. rHsGal-1 localizes at the site of injury and is found in intra and extracellular spaces.** A–B. Representative images of laser ablation assay with labeled 647rHsGal-1 (green) and FM1-43 Dye (red) with treatment time of 10min(A) and 48h(B). White arrows indicate site of injury. C. Quantification of the average ratio of corrected total cell 647rHsGal-1 fluorescence between inside and outside of the cell. D-I. Confocal images of A/J [-/-] myoblasts/myotubes treated and differentiated with 0.11 µM rHsGal-1 at varying time points: 10min (D), 4 h (E), 8 h (F), 24 h (G), 48 h (H), with labeled rHsGal-1(647rHsGal-1) (green) showing nucleus (blue) and membrane (red) identifying critical structures pertaining to rHsGal-1 location. 1 and 2 represents vesicle localization (I). Zoom: rHsGal-1 encapsulated in vesicles.

## Proteomic analysis shows rHsGal-1 increased expression of muscle development proteins in dysferlin-deficient myotubes

Lysates from WT, A/J$^{-/-}$ NT and A/J$^{-/-}$ rHsGal-1 treated myotubes were analyzed using shotgun proteomics and Gene Ontology Slim (GO-Slim) bioinformatics [41]. A heat map was generated using the top 20 upregulated and 20 downregulated proteins relative to average protein expression levels from all treatment groups (Fig 6A). Results showed an overlap of upregulated protein in the WT and rHsGal-1 treatment group compared to average expression (Fig 6A). We performed a GO-Slim analysis on all peptides with a fold change greater than 2 from NT to examine differences between NT and rHsGal-1 treatment proteome. Within this group, we examined the following GO-terms based on LGMD2B pathology that returned to WT levels after a 48 h treatment: 55% involved cellular component organization (GO: 0016043), 27.4% in cell and muscle differentiation (GO: 0030154 and GO:0043592), 13% in cell migration (GO: 0016477), and 4.3% inflammatory response (GO: 0006954) (Fig 6B and 6C).

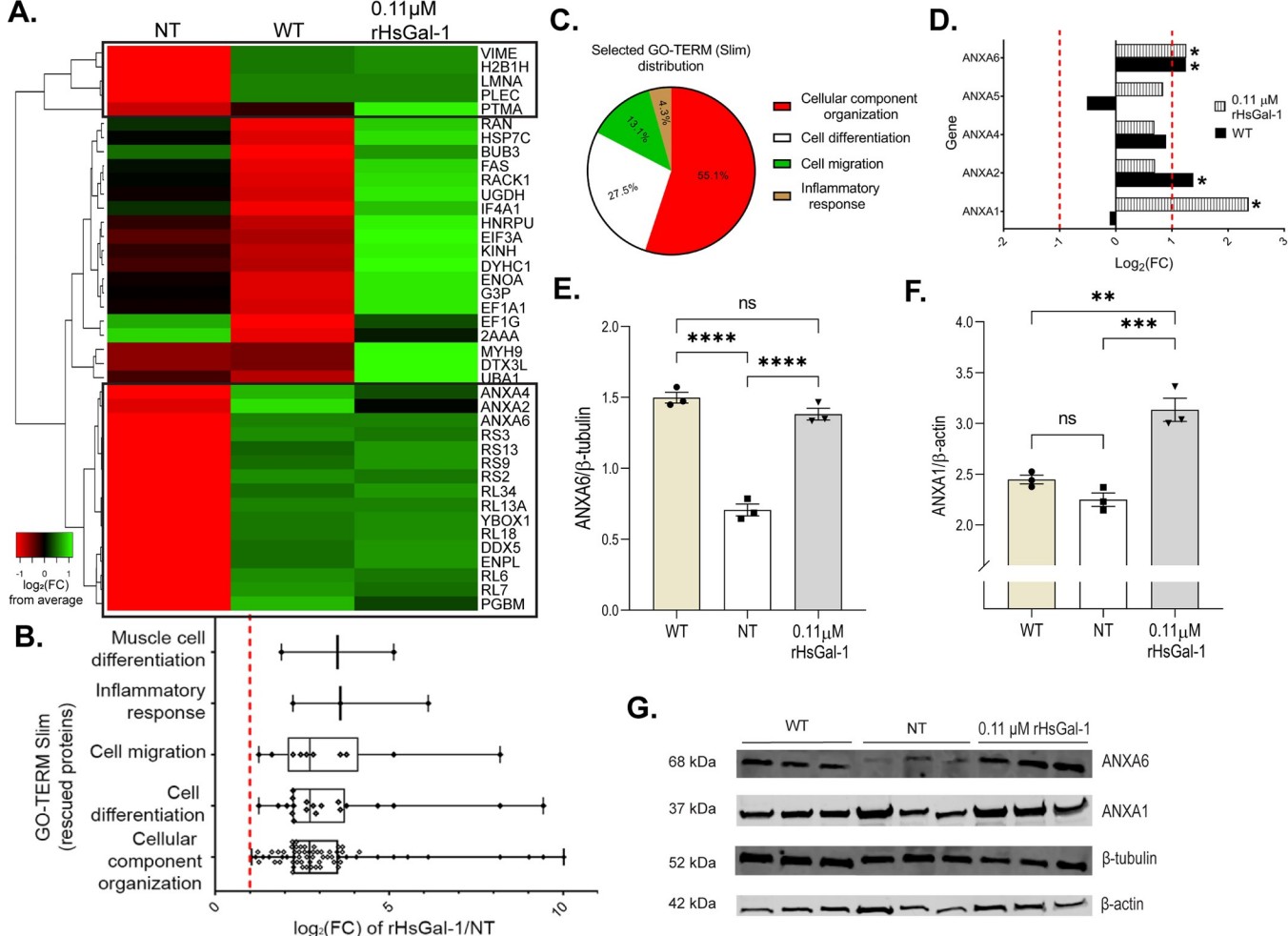

**Fig 6. Shotgun proteomic analyses (MS/MS) of AJ WT, AJ$^{-/-}$ NT and 0.11 µM rHsGal-1 treated myotubes.** A. Heatmap of top 20 and bottom 20 expressed proteins for WT myotubes based on Log$_2$ fold change (FC) from the mean value all conditions. Box denotes cluster of proteins that had similar FC values between WT and rHsGal-1 treatment. B. GO-Term Slim analysis of rescued proteins. C. Relative amount of proteins found in the selected GO-Terms(B). D. Log$_2$FC of Annexin family proteins (calculated based on WT and Gal-1 FC from NT). E. Quantification of ANXA6 after 48h differentiation in WT, NT, and 0.11 µM rHsGal-1 treated myotubes. F. Quantification of ANXA1 after 48h in WT, NT, and 0.11 µM rHsGal-1 treated myotubes. G. Western blot images of ANXA1 and ANXA6. p values were measured by 2-way ANOVA multiple comparison test and indicated by *p< 0.05, **p< 0.01, and ***p< 0.001. Error bars represent SEM. n = 3. All proteins were preselected based on protein ID significance values of ≥20% (p≤ 0.01, FDR≤0.05).

The annexin superfamily of proteins (ANXA) is important in many myogenic processes [42–44]. When we compared the proteomic results of WT and 0.11μM rHsGal-1 treated myotubes using their fold changes from the NT myotubes, *ANXA1*, *ANXA2*, *ANXA4*, and *ANXA6* were identified as being in the top 20 upregulated proteins with treatment (Fig 6A). The increased protein expression was confirmed via western blot analysis for *ANXA1* and *ANXA6* (Figs 6D and 5G). This increase was not seen in WT cells treated with rHsGal-1 (S4 Fig). Proteomic analysis supports the results that rHsGal-1 is a multifunctional protein capable of improving muscle development in LGMD2B.

Lysates from WT, A/J$^{-/-}$ NT, and rHsGal-1 treated myotubes were also analyzed using kinetic proteomics to determine protein turnover rate. Deuterium incorporation was measured at day 0, 3, 5, and 28 to determine the fraction of each protein that was newly synthesized. *ANXA1* and *ANXA6* from rHs-Gal-1 treated myotubes had increased turnover rate as compared to NT myotubes. Together, the upregulation and increased turnover rate shows that *ANXA1* and *ANXA6* are synthesized more with rHs-Gal-1 treatment (S5 Fig).

## Discussion

Current therapeutic options for LGMD2B are chiefly palliative in nature and do not present a significant quality of life benefit sought for by patients and their families. Steroid treatment to reduce chronic inflammation is negatively correlated with patient muscle strength and poses significant negative side effects [45]. Therefore, the need for developing an effective long-term treatment is imperative. Prior research using DMD murine models identified Gal-1 as an efficacious treatment in reducing disease symptoms [15]. Although DMD and LGMD2B have different genetic etiologies, both diseases converge in their disease pathologies, leading to decreased myogenic potential and aberrant repair. Here, we demonstrate that the ability of rHsGal-1 to improve myogenic factors and membrane repair reflects its therapeutic potential to decrease disease pathologies in LGMD2B.

Dysferlin-deficient muscle cells show a marked decrease in myogenic potential [46, 47]. After treatment with rHsGal-1, the expression of myogenic transcription factors reveals that dysferlin-deficient cells are committing to a myogenic lineage and maturing faster than non-treated dysferlin-deficient cells. The removal of treatment after a 10min rHsGal-1 followed by 72h differentiation was not sufficient to induce differences in myogenic, however, continuous 72h rHsGal-1 treatment coincided with increases in middle and late-stage markers. These results coupled with the formation of large multinucleated myotubes suggest that Gal-1 may help satellite cell commitment (Fig 2A and 2B and S2 Fig). During *in vivo* muscle development, self-alignment of myotubes during myogenesis is crucial to form healthy myofibers. Self-alignment is increased due to Gal-1 treatment in myotubes, indicating that an *in vivo* Gal-1 treatment has the potential for similar results (Fig 2F). We observed a concurrent upregulation of *LGALS1* transcript along with an increase in late-stage myogenic markers, suggesting a positive feedback loop with a 72h rHsGal-1 treatment, confirming the result seen in other models (Fig 1C, 1E, 1I and 1J) [32, 48]. One possible explanation for upregulated Gal-1 transcript is that increases in MyoD is known to robustly activate gene transcription possibly leading to eventual downstream Gal-1 transcription [49].

Since exogenous addition of Gal-1 does not increase proliferation [9], our findings suggest that Gal-1 treatment increased the rate of myogenesis in treated cells relative to their non-treated counterparts leading to increased myogenic fusion (Figs 1 and 2). *In vitro* proteomic analysis further supports these findings through the upregulated proteins involved in muscle cell differentiation, cellular differentiation, cell migration, and inflammatory response (Fig 6).

Ca$^{2+}$ sensitive C2 domains of dysferlin aid in plasma membrane resealing, a necessary process in myogenesis and wound healing [50]. In dysferlin-deficient myofibers and cells, this

process is compromised, which leads to diminished reseal capacity after injury, perpetuating LGMD2B disease pathology [40, 51, 52]. Kinetic laser injury results show that a 10min and 48h treatment improves membrane sealing; interestingly, the 10min treatment provided optimal membrane repair without upregulating myogenesis (Fig 3B, S2A–S2C Fig). However, this implies that rHsGal-1 induced improvement in myogenic potential alone cannot be responsible for the dramatic improvements in membrane sealing for this immediate result. Mechanical stabilization on the membrane due through the CRD of Gal-1 with known glycosylated membrane bound, EMC or yet to identified ligand could account for this action.

Differences in $Ca^{2+}$ involvement, along with temporal-spatial localization, helps narrow down possible mechanistic pathways responsible for observed increases in repair in a LGMD2B system. *Ex-vivo* results suggest that rHsGal-1 treatment improves membrane repair capacity in two different dysferlin-deficient strains of mice. Moreover, rHsgal-1 will not alter normal membrane repair functionality at this dose and is independent of $Ca^{2+}$ as we showed in A/J$^{-/-}$ and A/J$^{+/+}$ myotubes (Figs 3 and 4). One hypothesis that may offer explanation towards increase membrane repair capacity independent of $Ca^{2+}$ is rHsGal-1 treatment upregulates crucial membrane repair proteins such as ANXA1 and ANXA6 (Fig 6E–6G). The upregulation of *ANXA1* and *ANXA6* could also be attributed to differences in the rate of myogenesis since they are upregulated with differentiation.

The annexins regulate lipid binding, cytoskeletal reorganization, and muscle membrane repair cap formation [39, 43]. *ANXA1* and *ANXA6* have been observed to be involved in vesicle fusion, membrane resealing, muscle cell migration, and differentiation [43, 53]. Overexpression of ANXA6 promotes external blebbing and addition of exogenous ANXA6 increases membrane repair [44]. Our results show that rHsGal-1 treatment resulted in a significant increase of ANXA6 from NT in dysferlin-deficiency (Fig 6). Injury repair kinetics coupled with visualization of 647rHsGal-1 treatments show that Gal-1 accumulated at the site of laser injury in dysferlin-deficient myotubes. It is unlikely that these increased levels of ANXA are due to overall increased differentiation in LGMD2B models. Previous research has shown that exogenous Gal-1 treatment of both C2C12 cells and human fetal mesenchymal stem cells increase myogenic regulatory factors [54]. The lack of increase in ANXA levels in WT myotubes with treatment, lead us to conclude that this may be a specific to dysferlin-deficiency or an undefined interaction of ANXAs. This merits further inquiry to discover the mechanism responsible for Gal-1 interactions with specific substrates resulting in increased repair.

Our results indicate that the CRD of rHsGal-1 is an active structure required for therapeutic effect of Gal-1 in increasing membrane repair capacity (Fig 3). We believe the CRD provides mechanical stabilization to the membrane, aiding other cellular repair machinery to effectively mitigate damage and enhance repair. Mechanical stabilization also explains calcium-independent action for membrane repair and the difference in membrane repair seen at 10 min versus 48 hrs.

Healthy myogenic membranes after injury have a transient period of mobility and return to a state of low mobility [11, 55, 56]. One explanation for membrane destabilization and lack of proper repair in dysferlinopathy is an increase in lipid mobility [11]. Localization of labeled rHsGal-1 after 10min of treatment prior to injury is chiefly on the plasma membrane, whereas after a 48h treatment it appears as intracellular puncta, membrane bound puncta, and in lattice formation (Fig 5A and 5B). A previous study shows that after 1 h incubation Gal-1 localizes primarily intracellularly, which was confirmed in our 4 h 647rHsGal-1 images and quantification (Fig 5C) [57]. The greater accumulation of Gal-1 on the plasma membrane and increased repair capacity at 10min provides evidence that treatment may stabilize membrane associated proteins involved in repair enough to overcome the lack of dysferlin. Temporal-spatial images and fluorescent quantification provide evidence that by 8h, lattice structures are forming

which may further explain enhanced membrane stability and changes in protein interactions in LGMD2B (Fig 5C) [58, 59]. Furthermore, Gal-1 induced lattice formations appears to correlate with sites of cellular fusion (Fig 5 and S3 Fig, S2 Video). Future studies need to answer questions about localization and identification of rHsGal1 endogenous interactors or ligands at the sites of injury.

Cumulative results from this study provide evidence that rHsGal-1 may be a feasible protein therapeutic for LGMD2B by orchestrating a variety of changes that overcome intrinsic defects in myogenic functions. Increased connectivity observed in labeled rHsGal-1 may result in increased cellular signaling suggesting a potential mechanism for Gal-1 induced membrane repair that needs further investigation. Previous findings indicate localization of Gal-1 in the ECM [14]. The appearance of increased deposition of labeled rHsGal-1 in the extracellular space herein indicates that Gal-1 may increase skeletal muscle integrity in animal models of dysferlinopathy (Fig 5). These cumulative results support the hypothesis that the CRD mechanistically binds glycosylated membrane associated proteins, providing stability and overcoming inherent membrane destabilization due to lack of dysferlin. Although questions still remain about the nature of rHsGal-1 therapeutic mechanisms and systematic effects in more complex models of LGMD2B, these results provide evidence that Gal-1 is a viable therapeutic candidate in muscle diseases.

## Supporting information

**S1 Fig. Construct and purification of rHsGal-1.** A. rHsGal-1 construct which was inserted into pET-29b (+) vector. B. Coomassie Blue Stain of Gal-1. C. Ponceau S stain of Gal-1. D. Western blot image of Anti-Gal-1 at decreasing dosages. E. Western blot image of Anti-6x-His at decreasing dosages.
(TIF)

**S2 Fig. 10min treatment with rHsGal-1 does not influence myogenesis.** A. Western blot images of NT and 10min 0.11uM rHsGal-1 treated cells. B. Quantification of MYF5 expression. C. Quantification of MHC expression.
(TIF)

**S3 Fig. rHsGal-1 treatment increases membrane repair in A/J$^{-/-}$ myotubes independent of calcium.** A. Quantification of the change in fluorescent intensity in 0.11 μM rHsGal-1 treated A/J$^{-/-}$ myotube with or without extracellular $Ca^{2+}$ compared to NT myotubes supplemented or not with extracellular $Ca^{2+}$. B. Quantification of the change in fluorescent intensity in 0.11 μM rHsGal-1 treated A/J$^{-/-}$ myotube with or without EGTA compared to NT. C. Quantification of the change in fluorescent intensity in 0.11 μM rHsGal-1 treated A/J$^{-/-}$ myotubes with or without BAPTA-AM compared to NT. D. Representative images of NT and rHsGal-1 treated myotubes, with FM4-464 and Fluo-4AM, pre-injury, 1 s after injury, and 90s after injury. White arrows indicate site of injury. E. Quantification of Fluo-4 fluorescence within myotubes pre-injury, 1 s after injury, and 90s after injury. F. Representative images of A/J WT myotubes with FM4-64 and Fluo-4AM pre-injury and 1s, 35s, and 90s after injury. White arrows indicate site of injury. G. Quantification of change in Fluo-4AM fluorescence at injury. p values were measured by 2-way ANOVA multiple comparison test and indicated by $^*p< 0.05$, $^{**}p< 0.01$, $^{***}p< 0.001$, and $^{****}p< 0.0001$. Additionally, $\$p < .05$, and $\$ \$p < .01$ between NT and 0.11 μM rHsGal-1 0mM $Ca^{2+}$ treatment. Error bars represent SEM. n > 5 for each group.
(TIF)

**S4 Fig. rHsGal-1 does not affect WT levels of ANXA1/6.** A. Western blot images of A/J +/+ cells after no treatment (NT) or 48hr treatment with 0.11uM rHsGal-1. B. Quantification

of ANXA6 expression. C. Quantification of ANAX1 expression.
(TIF)

**S5 Fig. rHsGal-1 increases turnover rates of ANXA1 and ANXA6 compared to non-treated A/J$^{-/-}$ myotubes.** A–C. ANXA6 kinetic graphs quantifying the fraction of ANXA6 peptides incorporating D$_2$O over time in days. D–F. ANXA1 kinetic graphs quantifying the fraction of ANXA1 peptides incorporating D$_2$O over time in days. G. table with ANXA1 area (relative abundance) and turnover rate and its standard deviation. H. table with ANXA6 area (relative abundance) and turnover rate and its standard deviation.
(TIF)

**S1 Video. Laser injury assay.**
(AVI)

**S2 Video. Extracellular matrix rHsGal-1 lattice formation.**
(WMV)

**S3 Video. rHsGal-1 encapsulation in vesicles.**
(WMV)

**S1 Raw images.**
(TIF)

## Acknowledgments

The authors thank Dr. James Moody and Dr. John C. Price, Brigham Young University for technical assistance.

## Author Contributions

**Conceptualization:** Mary L. Vallecillo-Zúniga, Pam M. Van Ry.

**Data curation:** Mary L. Vallecillo-Zúniga.

**Formal analysis:** Mary L. Vallecillo-Zúniga, Pam M. Van Ry.

**Funding acquisition:** Pam M. Van Ry.

**Investigation:** Mary L. Vallecillo-Zúniga, Matthew F. Rathgeber, P. Daniel Poulson, Spencer Hayes, Jacob S. Luddington, Hailie N. Gill, Matthew Teynor, Braden C. Kartchner, Jonard Valdoz, Caleb Stowell, Ashley R. Markham, Pam M. Van Ry.

**Methodology:** Mary L. Vallecillo-Zúniga, Pam M. Van Ry.

**Project administration:** Pam M. Van Ry.

**Resources:** Sean Stowell, Pam M. Van Ry.

**Software:** Pam M. Van Ry.

**Supervision:** Pam M. Van Ry.

**Validation:** Mary L. Vallecillo-Zúniga.

**Visualization:** Mary L. Vallecillo-Zúniga, Jonard Valdoz.

**Writing – original draft:** Mary L. Vallecillo-Zúniga, Matthew F. Rathgeber, P. Daniel Poulson, Pam M. Van Ry.

**Writing – review & editing:** Mary L. Vallecillo-Zúniga, Matthew F. Rathgeber, P. Daniel Poulson, Spencer Hayes, Jacob S. Luddington, Hailie N. Gill, Matthew Teynor, Braden C. Kartchner, Jonard Valdoz, Caleb Stowell, Ashley R. Markham, Connie Arthur, Sean Stowell, Pam M. Van Ry.

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
