## [Decision Letter · Decision Letter 0]

12 Jun 2020

PONE-D-20-15679

Treatment with Galectin-1 Improves Myogenic Potential and Membrane Repair in Dysferlin-deficient Models

PLOS ONE

Dear Dr. Van Ry,

Thank you for submitting your manuscript to PLOS ONE. After careful consideration, we feel that it has merit but does not fully meet PLOS ONE’s publication criteria as it currently stands. The reviewers found your study very interesting but different points need to be addressed in order to strengthen the conclusions. Therefore, we invite you to submit a revised version of the manuscript that addresses all the points raised by the reviewers.

We look forward to receiving your revised manuscript.

Kind regards,

Antonio Musaro, Ph.D.

Academic Editor

PLOS ONE

Journal Requirements:

4. We note that you have a patent relating to material pertinent to this article. Please provide an amended statement of Competing Interests to declare this patent (with details including name and number), along with any other relevant declarations relating to employment, consultancy, patents, products in development or modified products etc. Please confirm that this does not alter your adherence to all PLOS ONE policies on sharing data and materials, as detailed online in our guide for authors http://journals.plos.org/plosone/s/competing-interests by including the following statement: "This does not alter our adherence to  PLOS ONE policies on sharing data and materials.” If there are restrictions on sharing of data and/or materials, please state these. Please note that we cannot proceed with consideration of your article until this information has been declared.

Additional Editor Comments (if provided):

Reviewers' comments:

Reviewer's Responses to Questions

**Comments to the Author**

1. Is the manuscript technically sound, and do the data support the conclusions?

Reviewer #1: Yes

Reviewer #2: Partly

2. Has the statistical analysis been performed appropriately and rigorously? 

Reviewer #1: Yes

Reviewer #2: I Don't Know

3. Have the authors made all data underlying the findings in their manuscript fully available?

Reviewer #1: Yes

Reviewer #2: Yes

4. Is the manuscript presented in an intelligible fashion and written in standard English?

Reviewer #1: Yes

Reviewer #2: Yes

5. Review Comments to the Author

Reviewer #1: This is a very interesting paper addressing an issue of muscle repair in dysferlin-deficient mice, which are modeled after the human Limb-girdle muscular dystrophy 2B (LGMD2B), which is a type of dysferlinopathy. Loss of dysferlin is known to affect calcium signaling and is associated with poor membrane repair, myogenesis, and muscle degeneration. The results presented here demonstrate that treatment of cells and ex-vivo muscle with a recombinant form of a carbohydrate-binding multivalent lectin human galectin-1 (rHsGal-1) both improves membrane repair and exhibits calcium-independent membrane repair in dysferlin-deficient and wild-type myotubes and myofibers. Mechanistically, the effect observed are demonstrated to occur through carbohydrate-dependent recognition and act through a pathway of myogenic potential and mechanical stabilization of the membrane.

Overall, this is a robust and interesting study that is well-done and offers novel insights into LGMD2B and even potential novel current possibilities.

Although the study is overall very well done, and well-written in a critical way, there are a few questions the authors should address.

1. The authors use the human galectin-1 protein in these studies instead of the murine galectin-1, and therefore should comment on their relatedness in terms of carbohydrate-biding specificity and the rationale for their choice.

2. Is the rHsGal-1 internalized by the cells and is it stable in the culture treatments? ‘

3. The authors show that treatment with rHsGal-1 increased the transcript and production of endogenous galectin-1? Is that protein secreted by the cells and does it contribute to the results?

4. Do mice lacking in galectin-1 have similar responses? Are the effects observed entirely from the exogenous rHsGal-1?

5. Do the effects observed, for example in Fig. 1-3, require continuous exposure to active rHsGal-1, e.g. if rHsGal-1 is added and then removed or blocked by addition of a hapten, are the effects not observed?

6. The localization of rHsGal-1 to sites of injury suggest special endogenous ligands are present there, so do the authors have any ideas about such ligands, and are they increased upon injury or simply ‘relocalized’ to the injury site?

Reviewer #2: This work is investigating the potential therapeutic role of galectin-1 for LGMD2B (dysferlin deficient) muscles.

1. Although Gal-1 treatment clearly shows increased myogenic potential of dysferlin deficient myotubes and increased membrane repair capacity, respectively, ‘increased membrane repair capacity by increasing myogenic potential’ (quote from abstract) is questionable. Because 10 minute treatment of Gal-1 in A/J -/- myotubes and 2hr treatment of Gal-1 in Bla/J and Dysf-/- myofibers recover membrane repair defect robustly, increasing myogenic potential, which would take more than 2 hours, would be hard to induce membrane repair. Author should provide a valid evidence showing connection between myogenic potential and membrane repair capacity or should disconnect between.

2. FM1-43 is a way of showing membrane closing time by repair. Calcium influx and phosphatidyl serine exposure (lactaherin or Annexin-V) also show membrane repair. Because Gal-1 shows calcium independent membrane repair, calcium imaging (membrane repair with Gal-1) is necessary to show.

3. Gal-1 48h treatment cause different level of muscle differentiation. So untreated A/J-/- cells are myoblast-like cells and treated ones are myotube-like cells. Membrane repair level could be different between myoblast and myotubes because many membrane repair proteins including annexins are increased during myogenesis (https://pubmed.ncbi.nlm.nih.gov/23277424/). Therefore, author should be careful to use Gal-1 48 hr treated cells to argue the recovery of membrane repair capacity. It could be a result of difference between myoblast and myotubes. Fig 6 has similar issues. Annexin protein increasing could be a result of different level of differentiation rather than induction by gal-1. If author show increase of Annexin proteins in WT cells by Gal-1, assuming Gal-1 doesn’t induce robust differentiation in WT, Annexin protein increase by Gal-1 can be accepted.

4. Fig 5 seems weak to be a main figure. It provide more detailed info., but basically same info. with Fig 4. Or, Fig 5 should be quantified to show Gal-1 movement thru time course.

Minor

1. Show individual data points in the bar graph.

2. Indicate biological replicate and sample numbers in the figure legends. For example, Fig 2G, WT (n=30), NT (n=34)… should be ‘30 WT myotubes from n experiments’. Since cell lines were used, it is important to mention how many independent experiments are performed to get data. Please note independent experiments should be performed in different day. If cells are plated into 3 wells in a plate and treated by same solution at same time, it is a technical triplicate not 3 independent experiments.

3. Add proper control for Fig 3. For Fig 3B and 3D, WT value could be helpful to readers to understand the degree of rescue by Gal-1.

4. Provide logic to use injured fiber in fig 7D. If possible, add uninjured WT fibers (with without gal-1) as control.

5. This could be out of scope comments but author could try inject Gal-1 in Dysf-/- or Bla/J mice to show reduced Creatin kinase levels (may be after exercise?) as prove of in vivo therapeutic effect of gal-1.

6. PLOS authors have the option to publish the peer review history of their article (what does this mean?). If published, this will include your full peer review and any attached files.

Reviewer #1: No

Reviewer #2: No

---

## [Author Response · Author response to Decision Letter 0]

28 Jul 2020

Reviewer #1:

1. The authors use the human galectin-1 protein in these studies instead of the murine galectin-1, and therefore should comment on their relatedness in terms of carbohydrate-biding specificity and the rationale for their choice.

We added additional language in the introduction to clarify the homology between mouse and human Galectin-1 (line 63-67)

“Galectin-1 (Gal-1) is a small, non-glycosylated protein encoded by the LGALS1 gene with a carbohydrate recognition domain (CRD) which is highly conserved between all mammals with an 88% homology. [14-18] Mouse and human Gal-1 have minor structural differences, but the carbohydrate recognition residues are 100% conserved. Mice lacking Gal-1 showed a reduction in myoblast fusion and muscle regeneration.”

Our rational for the choice of Galectin-1 was added in the introduction (lines 70-71).

“Since previous research using rHsGal-1 was similar to those reported using recombinant mouse Gal-1 in a DMD mouse model, we chose to use rHsGal-1 in our study.”

In addition to this comment some of my work in a previous lab investigated the pharmacokinetics, pharmacodynamics and toxicity of rHsGal-1 in a DMD mouse model (see reference 58).

2. Is the rHsGal-1 internalized by the cells and is it stable in the culture treatments? 

Yes. Thank you for this question. It allowed us to address an important aspect of this treatment. Galectin-1 appears to be stable in this cell model for at least up to the 72-hour timepoint. We did not test longer. We addressed this comment on lines 426-428.

“However, after a 48h treatment there was minimal rHsGal-1 localized on the myotube membrane and instead formed puncta within the cytosol (Fig 4B and 4H), further indicating the stability of the exogenous Gal-1 within these cells.”

3. The authors show that treatment with rHsGal-1 increased the transcript and production of endogenous galectin-1? Is that protein secreted by the cells and does it contribute to the results? 

The answer to both of these questions is yes.

4. Do mice lacking in galectin-1 have similar responses? Are the effects observed entirely from the exogenous rHsGal-1?

We added language addressing Galectin-1 knockout mice (line 66-67).

“Mice lacking Gal-1 showed a reduction in myoblast fusion and muscle regeneration.”

It is our hypothesis that the results stem from exogenous rHsGal-1. Although the addition of exogenous Gal-1 is known to cause further upregulation of transcript and protein levels of endogenous Gal-1. Thus, the results could also be attributed to the increase of endogenous Gal-1. This is stated in lines 539-544.

“We observed a concurrent upregulation of LGALS1 transcript along with an increase in late-stage myogenic markers, suggesting a positive feedback loop with a 72h rHsGal-1 treatment, confirming the result seen in other models (Figs 1C, 1E, 1I, and 1J). [48, 49] One possible explanation for upregulated Gal-1 transcript is that increases in MyoD is known to robustly activate gene transcription possibly leading to eventual downstream Gal-1 transcription.[50]”

5. Do the effects observed, for example in Fig. 1-3, require continuous exposure to active rHsGal-1, e.g. if rHsGal-1 is added and then removed or blocked by addition of a hapten, are the effects not observed?

Thank you for the suggestion. To address this question, we treated confluent myoblasts with rHsGal-1 for 10 min, removed the treatment and allowed differentiation for an additional 72 hours (see supplemental figure S2, lines 294-296)

“The removal of 0.11μM rHsGal-1 after a 10min treatment in A/J -/- myotubes followed by 72 hours in differentiation media show no significant difference in Myf5 or MHC expression when compared to NT (Fig S2A-S2C)”.

(lines 532-534).

“The removal of treatment after a 10min rHsGal-1 followed by 72h differentiation was not sufficient to induce differences in myogenic, however, continuous 72h rHsGal-1 treatment coincided with increases in middle and late-stage markers.”

In figure 3C the carbohydrate recognition domain is blocked by using lactose which results in no increased repair capacity (lines 377-379, fig 3C). 

6. The localization of rHsGal-1 to sites of injury suggest special endogenous ligands are present there, so do the authors have any ideas about such ligands, and are they increased upon injury or simply ‘relocalized’ to the injury site?

We do not know the full answer to this question. We have postulated it could be interactions with Annexins, but the Gal-1 interactome/ligands in muscular dystrophy models are not yet known. We believe this set of experiments are beyond the scope of this paper, but are currently designing experiments to answer this question for a future paper. We did add additional language stating this in lines 603-604, 570-583 below.

“Future studies need to answer questions about localization and identification of rHsGal1 endogenous interactors or ligands at the sites of injury.”

The annexins regulate lipid binding, cytoskeletal reorganization, and muscle membrane repair cap formation.[39, 43] ANXA1 and ANXA6 have been observed to be involved in vesicle fusion, membrane resealing, muscle cell migration, and differentiation.[43, 59] Overexpression of ANXA6 promotes external blebbing and addition of exogenous ANXA6 increases membrane repair.[44] Our results show that rHsGal-1 treatment resulted in a significant increase of ANXA6 from NT (Fig 5). These increased levels of ANXA may be due to overall increased differentiation in LGMD2B models, but the lack of increase in ANXA levels in WT lead us to conclude that there may be a specific interaction. Injury repair kinetics coupled with visualization of 647rHsGal-1 treatments show that Gal-1 accumulated at the site of laser injury in dysferlin-deficient myotubes. This merits further inquiry to discover the mechanism responsible for Gal-1 interactions with specific substrates resulting in increased repair.”

Reviewer #2: 

Major Points

1. This work is investigating the potential therapeutic role of galectin-1 for LGMD2B (dysferlin deficient) muscles.

Although Gal-1 treatment clearly shows increased myogenic potential of dysferlin deficient myotubes and increased membrane repair capacity, respectively, ‘increased membrane repair capacity by increasing myogenic potential’ (quote from abstract) is questionable. Because 10 minute treatment of Gal-1 in A/J -/- myotubes and 2hr treatment of Gal-1 in Bla/J and Dysf-/- myofibers recover membrane repair defect robustly, increasing myogenic potential, which would take more than 2 hours, would be hard to induce membrane repair. Author should provide a valid evidence showing connection between myogenic potential and membrane repair capacity or should disconnect between.

Thank you for this insightful comment. You’re completely correct and we carefully disconnected myogenic potential and membrane repair capacity for the different treatment time points. lines 587-589 and lines 33-36

“Mechanical stabilization also explains calcium-independent action for membrane repair and the difference in membrane repair seen at 10 min versus 48 hrs.” 

We also change this original statement in the abstract:

“These results suggest that rHsGal-1 improves myogenic potential and membrane repair through mechanical stabilization of the membrane.” To “Improvements in membrane repair after only a 10min rHsGal-1treatment suggest mechanical stabilization of the membrane due to interaction with glycosylated membrane bound, ECM or yet to be identified ligands through the CDR domain of Gal-1.”

To further address this point, we treated confluent myoblasts with rHsGal-1 for 10 min, removed the treatment and allowed differentiation for an additional 72 hours (see supplemental figure S2, lines 294-296). There were no changes in myogenic potential with the removal of treatment.

“The removal of 0.11μM rHsGal-1 after a 10min treatment in A/J -/- myotubes followed by 72 hours in differentiation media show no significant difference in Myf5 or MHC expression when compared to NT (Fig S2A-S2C)”.

(lines 532-534).

“The removal of treatment after a 10min rHsGal-1 followed by 72h differentiation was not sufficient to induce differences in myogenic, however, continuous 72h rHsGal-1 treatment coincided with increases in middle and late-stage markers.”

2. FM1-43 is a way of showing membrane closing time by repair. Calcium influx and phosphatidyl serine exposure (lactaherin or Annexin-V) also show membrane repair. Because Gal-1 shows calcium independent membrane repair, calcium imaging (membrane repair with Gal-1) is necessary to show.

Thank you for the suggestion to add calcium imaging. We monitored calcium influx during laser injury using Fluo-4 AM Calcium fluorescent imaging. The data from this experiment is in Figure S3. I have included the additional comments that were added to the results section (lines 407-412).

“We saw that rHsGal-1 treatment increases membrane repair and mitigates effects of dysferlin-deficiency in the presence of both intracellular and extracellular calcium chelators (Figs S3B and S3C). Calcium imaging using Fluo-4AM also revealed no increase in Ca2+ accumulation at site of injury in A/J -/- 0.11μM rHsGal-1 treated and NT myotubes, but did find an increase in Ca2+ accumulation at the site of injury in A/J WT myotubes (Figs S3D –S3G).” 

3. Gal-1 48h treatment cause different level of muscle differentiation. So untreated A/J-/- cells are myoblast-like cells and treated ones are myotube-like cells. Membrane repair level could be different between myoblast and myotubes because many membrane repair proteins including annexins are increased during myogenesis (https://pubmed.ncbi.nlm.nih.gov/23277424/). Therefore, author should be careful to use Gal-1 48 hr treated cells to argue the recovery of membrane repair capacity. It could be a result of difference between myoblast and myotubes.

By separating increases in myogenic potential of which myogenesis is a part from membrane repair capacity, we have addressed some of the confusion with this point. Only myotubes with 3 or > nuclei were used in laser injury repair assays. We added additional language to argue that the increased repair capacity at 10min is likely due to increased interactions with glycosylated membrane bound or ECM proteins through the CRD of Gal-1and not due to increased myogenesis. We also added a new Fig S2A. While treatments for longer periods show both increased myogenic rate and increased membrane repair capacity. 

 Fig 6 has similar issues. Annexin protein increasing could be a result of different level of differentiation rather than induction by gal-1. If author show increase of Annexin proteins in WT cells by Gal-1, assuming Gal-1 doesn’t induce robust differentiation in WT, Annexin protein increase by Gal-1 can be accepted.

We added language addressing the possible increased levels of ANXA could be due to differences in differentiation (lines 567-569). 

“The upregulation of ANXA1 and ANXA6 could also be attributed to differences in the rate of myogenesis since they are upregulated with differentiation.”

We also added figure S4. Although the figure does not show an increase of Annexin. We address these concerns in the discussion with the following quote: (lines 565-582) 

“One hypothesis that may offer explanation towards increase membrane repair capacity independent of Ca2+ is rHsGal-1 treatment upregulates crucial membrane repair proteins such as ANXA1 and ANXA6 (Figs 5E-5G). The upregulation of ANXA1 and ANXA6 could also be attributed to differences in the rate of myogenesis since they are upregulated with differentiation.

The annexins regulate lipid binding, cytoskeletal reorganization, and muscle membrane repair cap formation.[39, 43] ANXA1 and ANXA6 have been observed to be involved in vesicle fusion, membrane resealing, muscle cell migration, and differentiation.[43, 59] Overexpression of ANXA6 promotes external blebbing and addition of exogenous ANXA6 increases membrane repair.[44] Our results show that rHsGal-1 treatment resulted in a significant increase of ANXA6 from NT in dysferlin-deficiency (Fig 5). It is unlikely that these increased levels of ANXA are due to overall increased differentiation in LGMD2B models. Previous research has shown that exogenous Gal-1 treatment of both C2C12 cells and human fetal mesenchymal stem cells increase myogenic regulatory factors[Fisk 2006]. The lack of increase in ANXA levels in WT myotubes with treatment, lead us to conclude that this may be a specific to dysferlin-deficiency or an undefined interaction of ANXAs.”

4. Fig 5 seems weak to be a main figure. It provide more detailed info., but basically same info. with Fig 4. Or, Fig 5 should be quantified to show Gal-1 movement thru time course.

Thank you for the suggestion, we combined figure 4 and 5 as suggested and quantified the movement of labeled Gal-1 over time.

Minor Points

1. . Show individual data points in the bar graph. Addressed, we changed all our bar graphs to show individual data points.

2. Indicate biological replicate and sample numbers in the figure legends. For example, Fig 2G, WT (n=30), NT (n=34)… should be ‘30 WT myotubes from n experiments’. Since cell lines were used, it is important to mention how many independent experiments are performed to get data. Please note independent experiments should be performed in different day. If cells are plated into 3 wells in a plate and treated by same solution at same time, it is a technical triplicate not 3 independent experiments. Thank you this is a good clarification. We added the suggested notations to the figure legends.

3. Add proper control for Fig 3. For Fig 3B and 3D, WT value could be helpful to readers to understand the degree of rescue by Gal-1. We added the WT values to both 3B and 3D. 

4. Provide logic to use injured fiber in fig 7D. If possible, add uninjured WT fibers (with without gal-1) as control. Thank you for pointing out the ambiguity, we added additional language to clarify that pre-injury images are uninjured fibers and added WT data to the graphs (lines 201-204).

“A TCS SP2 two-photon confocal scanning microscope (Leica) was used to injure the membrane of a myotube or myofiber and images were taken before and after the injury event. Pre-injury images depict uninjured myofibers.”

5. This could be out of scope comments but author could try inject Gal-1 in Dysf-/- or Bla/J mice to show reduced Creatin kinase levels (may be after exercise?) as prove of in vivo therapeutic effect of gal-1. This is great suggestion, but we believe it is beyond the scope of this paper. We are working on this data, but it requires multiple doses and frequency regiments to establish efficacious dose.

---

## [Decision Letter · Decision Letter 1]

12 Aug 2020

Dear Dr. Van Ry,

After careful consideration, we feel that it has merit but, in order to strengthen the conclusions, we invite you to submit a revised version of the manuscript that addresses the minor points raised during the review process.

We look forward to receiving your revised manuscript.

Kind regards,

Antonio Musaro, Ph.D.

Academic Editor

PLOS ONE

Reviewers' comments:

Reviewer's Responses to Questions

**Comments to the Author**

1. If the authors have adequately addressed your comments raised in a previous round of review and you feel that this manuscript is now acceptable for publication, you may indicate that here to bypass the “Comments to the Author” section, enter your conflict of interest statement in the “Confidential to Editor” section, and submit your "Accept" recommendation.

Reviewer #1: (No Response)

Reviewer #2: All comments have been addressed

2. Is the manuscript technically sound, and do the data support the conclusions?

Reviewer #1: (No Response)

Reviewer #2: Yes

3. Has the statistical analysis been performed appropriately and rigorously? 

Reviewer #1: (No Response)

Reviewer #2: Yes

4. Have the authors made all data underlying the findings in their manuscript fully available?

Reviewer #1: (No Response)

Reviewer #2: Yes

5. Is the manuscript presented in an intelligible fashion and written in standard English?

Reviewer #1: (No Response)

Reviewer #2: Yes

6. Review Comments to the Author

Reviewer #1: This is a very interesting paper addressing an issue of muscle repair in dysferlin-deficient mice, which are modeled after the human Limb-girdle muscular dystrophy 2B (LGMD2B), which is a type of dysferlinopathy. Loss of dysferlin is known to affect calcium signaling and is associated with poor membrane repair, myogenesis, and muscle degeneration. The results presented here demonstrate that treatment of cells and ex-vivo muscle with a recombinant form of a carbohydrate-binding multivalent lectin human galectin-1 (rHsGal-1) both improves membrane repair and exhibits calcium-independent membrane repair in dysferlin-deficient and wild-type myotubes and myofibers. Mechanistically, the effect observed are demonstrated to occur through carbohydrate-dependent recognition and act through a pathway of myogenic potential and mechanical stabilization of the membrane.

The authors have adequately addressed all the major concerns raised in the initial review.

Reviewer #2: One minor comment is that rearrange Fig 6. If Fig 6 is moved after Fig 3 or 4, story would be stronger since it is showing that enhanced membrane repair of Gal-1 treated Dysf-/-myofibers, which is more physiologically relevant than cultured myotubes.

7. PLOS authors have the option to publish the peer review history of their article (what does this mean?). If published, this will include your full peer review and any attached files.

Reviewer #1: No

Reviewer #2: No

---

## [Author Response · Author response to Decision Letter 1]

13 Aug 2020

Reviewer #2:

1. One minor comment is that rearrange Fig 6. If Fig 6 is moved after Fig 3 or 4, story would be stronger since it is showing that enhanced membrane repair of Gal-1 treated Dysf-/-myofibers, which is more physiologically relevant than cultured myotubes.

As suggested, we rearranged the manuscript so that Fig 6 is now Fig 4. All reference to figures in the manuscript have been updated so they coincide with the new figure numbering. Additionally, we have rearranged all relevant paragraphs to strengthen the story.

---

## [Editor Report · Decision Letter 2]

18 Aug 2020

Treatment with Galectin-1 Improves Myogenic Potential and Membrane Repair in Dysferlin-deficient Models

PONE-D-20-15679R2

Dear Dr. Van Ry,

We’re pleased to inform you that your manuscript has been judged scientifically suitable for publication and will be formally accepted for publication once it meets all outstanding technical requirements.

Kind regards,

Antonio Musaro, Ph.D.

Academic Editor

PLOS ONE

Additional Editor Comments (optional):

The authors adequately addressed the points raised by the Reviewers
---

## [Editor Report · Acceptance letter]

20 Aug 2020

PONE-D-20-15679R2 

Treatment with galectin-1 improves myogenic potential and membrane repair in dysferlin-deficient models 

Dear Dr. Van Ry:

I'm pleased to inform you that your manuscript has been deemed suitable for publication in PLOS ONE. Congratulations! Your manuscript is now with our production department. 

Kind regards, 

on behalf of

Dr. Antonio Musaro 

Academic Editor

PLOS ONE